# Synthesis, In Vitro α-Glucosidase Inhibitory Activity and Molecular Docking Study of New Benzotriazole-Based Bis-Schiff Base Derivatives

**DOI:** 10.3390/ph16010017

**Published:** 2022-12-22

**Authors:** Imran Khan, Wajid Rehman, Fazal Rahim, Rafaqat Hussain, Shoaib Khan, Srosh Fazil, Liaqat Rasheed, Muhammad Taha, Syed Adnan Ali Shah, Magda H. Abdellattif, Thoraya A. Farghaly

**Affiliations:** 1Department of Chemistry, Hazara University, Mansehra 21120, Pakistan; 2Department of Chemistry, University of Poonch Rawalakot, Azad Kashmir 12350, Pakistan; 3Department of Clinical Pharmacy, Institute for Research and Medical Consultations (IRMC), Imam Abdul Rahman Bin Faisal University, P.O. Box 31441, Dammam 34212, Saudi Arabia; 4Department of Chemistry, College of Sciences, Taif University, P.O. Box 11099, Taif 21944, Saudi Arabia; 5Department of Chemistry, Faculty of Applied Science, Umm Al-Qura University, P.O. Box 715, Makkah Almukkarramah 24382, Saudi Arabia

**Keywords:** synthesis, α-glucosidase, benzotriazole, bis-Schiff base, structure activity relationship, molecular docking

## Abstract

This study was carried out to synthesize benzotriazole-based bis-Schiff base scaffolds (**1**–**20**) and assess them in vitro for *α*-glucosidase inhibitory potentials. All the synthetics analogs based on benzotriazole-based bis-Schiff base scaffolds were found to display an outstanding inhibition profile on screening against the *α*-glucosidase enzyme. The synthetic scaffolds showed a varied range of inhibition profiles having IC_50_ values ranging from 1.10 ± 0.05 µM to 28.30 ± 0.60 µM when compared to acarbose as a standard drug (IC_50_ = 10.30 ± 0.20 µM)**.** Among the series, fifteen scaffolds **1**–**3**, **5**, **6**, **9**–**16**, **18**–**20** were identified to be more potent than standard acarbose, while the five remaining scaffolds **4**, **7**, **8**, **16,** and **17**, also showed potency against the *α*-glucosidase enzyme but were found to be less potent than standard acarbose. The structure of all the newly synthesized scaffolds was confirmed using different spectroscopic techniques such as HREI-MS and ^1^H- and ^13^C- NMR spectroscopy. To find a structure-activity relationship, molecular docking studies were carried out to understand the binding mode of the active inhibitors with the active sites of the enzyme and the results supported the experimental data.

## 1. Introduction

Diabetes mellitus (DM) is a well-known, progressive endocrine disorder associated with increased morbidity and mortality, as well as high healthcare costs. There were approximately 171 million cases of DM in 2000, and this number is expected to more than double over the next 25 years, reaching 366 million by 2030 [1,2]. This rising trend has prompted the development of new therapeutic agents as a serious global medical concern. Due to deficiencies in insulin production or action, DM is characterized by hyperglycemia and changes in the metabolisms of carbohydrates, proteins, and lipids [3]. Postprandial hyperglycemia is a prominent defect that occurs early in diabetes and may lead to various secondary complications, including elevated risk for cardiovascular diseases [4], atherosclerosis, cataracts, retinopathy, neuropathy, nephropathy, and impaired wound healing [5]. α-glucosidase (EC 3.2.1.20), an enzyme found in the epithelial mucosa of the small intestine that cleaves the glycosidic bonds in complex carbohydrates to release absorbable monosaccharide, aids in the rapid absorption of carbohydrates, which causes elevated blood glucose levels when consumed [6].

One therapeutic approach for treating diabetes involves controlling postprandial hyperglycemia by inhibiting the α-glucosidase in the digestive tract, delaying and prolonging the overall carbohydrate digestion time. Slowing carbohydrate digestion should reduce the rate of glucose absorption and consequently prevent spikes in postprandial blood glucose and insulin levels [7,8]. Using α-glucosidase inhibitors has become a promising therapeutic strategy for reducing the risks of diabetes and other carbohydrate-mediated diseases, including hyperlipoproteinemia and obesity [9,10,11,12,13,14,15].

Benzotriazole scaffolds were reported as key precursors in designing and developing biologically more-active drugs and were known to have a broad range of biological activities such as antibacterial [16], antifungal [17,18], antihistaminic, anti-adrenergic, and analgesic [19], anti-cancer [20] and anti-convulsant activities [21].

Viewing the biological importance of benzotriazole [22] and bis-Schiff base analogs [23], herein this study we plan to synthesize a library of new benzotriazole-bearing bis-Schiff base derivatives as potent α-glucosidase inhibitors. Therefore, in the current study, we synthesized unreported benzotriazole-bearing bis-Schiff base analogs. We believe such compounds are more lipophilic and will increase activity and easily pass through the cell wall (Figure 1).

## 2. Result and Discussion

### 2.1. Chemistry

In the first step, the pre-synthesized 1,2,3-benzotriazole (**I**) was treated with various phenacyl bromide (**II**) concentrations in ethanol and triethylamine and the resulting mixture of reaction was permitted to put on reflux with gentle stirring until the formation of 2-(1H-benzo[d][1,2,3]triazol-1-yl)-1-phenylethan-1-one substrate (**III**) was complete (the conversion was monitored by TLC, reflux, 3 hrs). After completion of the reaction, the residue was stored overnight on cooling for crystallization of the desired substrate (**III**)**.** In the next step, the carbonyl residue of the substrate (**III**) underwent condensation with hydrated hydrazine in acetic acid, and the resulting mixture was stirred for 3 hrs to achieve the formation of (E)-1-(2-hydrazono-2-phenylethyl)-1H-benzo[d][1,2,3]triazole (**IV**)**,** which was treated with various benzaldehyde concentrations in acetic acid to afford the synthesis of 1,2,3-benzotriazole-basedbis-Schiff base derivatives (**1**–**20**) (Figure 1).

### 2.2. In Vitro α-Glucosidase Inhibition Profile

To treat various diseases, the discovery and development of more potent drugs through the inhibition of enzymes has received much attention from medicinal chemists in the past few years. Benzotriazole-based bis-Schiff base scaffolds (**1**–**20**) were synthesized and then investigated in vitro for *α*-glucosidase inhibitory potentials. All the synthetic analogs based on benzotriazole-based bis-Schiff base scaffolds were found to display outstanding inhibition profiles on screening against the *α*-glucosidase enzyme. The synthetic scaffolds showed a varied range of inhibition profiles having IC_50_ values ranging from 1.10 ± 0.05 µM to 28.30 ± 0.60 µM when compared to acarbose as a standard drug (IC_50_ = 10.30 ± 0.20 µM) (Table 1). Among the series, fifteen scaffolds **1**–**3**, **5**, **6**, **9**–**16**, **18**–**20** were identified to be more potent than standard acarbose, while the five remaining scaffolds **4**, **7**, **8**, **16**, and **17** also showed potency against the *α*-glucosidase enzyme but were found to be less potent than standard acarbose.

#### Structure-Activity Relationship (SAR) for α-Glucosidase Enzyme

Structure-activity relationship (SAR) studies suggested that the variation in number/s, position, and nature of substituents around both aryl parts **‘B’** and **‘C’** of the benzotriazole-based bis-Schiff base scaffolds (**1**–**20**) greatly affect the inhibitory potentials of the *α*-glucosidase enzyme.

Compounds (**1**, **9**, **10,** and **13**), bearing tri-substitutions such as di-Cl at the 3,5-position, –OH group at the 2-position of aryl part ‘C’ and a variety of other groups such as –OCH_3_, di-Cl and –NO_2_ at various position of aryl part ‘B’, were shown to display considerable inhibitory potentials against the *α*-glucosidase enzyme. Scaffold (**9**), which has a –NO_2_ moiety at the 3-position of aryl part ‘B’ and tri-substitutions –OH and di-Cl moieties present at the 2,3,5-position of aryl part ‘C’, was identified as the most active inhibitor of the *α*-glucosidase enzyme with nine-fold greater potency than standard acarbose. However, scaffold (**10**) showed somewhat less potency than scaffold (**9**) even though it also bears a substituent of the same nature; a slight difference in inhibitory potentials may be attributed to an alteration in the position of the –NO_2_ moiety on aryl part ‘B’. The enhanced inhibitory potentials of both these analogs (**9**) and (**10**) against *α*-glucosidase may perhaps be due to greater numbers of e-withdrawing groups (di-Cl and –NO_2_) attached to both aryl parts ‘B’ and ‘C’, which make both aryl rings more susceptible for interactions with the active residues of the *α*-glucosidase enzyme. Moreover, attached substituents such as –OH and –NO_2_ groups (capable of interactions through H-bonding with the active site of targeted enzyme) also enhanced the enzymatic activity of scaffolds (**9**) and (**10**). Compound (**1**) displayed a four-fold lower potency than compound (**9**). This decline in the inhibition profile of analog (**1**) was due to the replacement of the –NO_2_ moiety with a –OCH_3_ group, showing that the nature of attached substituents greatly affects the potency of the synthesized scaffolds against the *α*-glucosidase enzyme. It was also observed that replacement of the –OCH_3_ group of compound (**1**) with a hydrogen atom followed by subsequent addition of di-Cl moieties at the 2,4-position of aryl part ‘B’ enhanced the *α*-glucosidase activity by three-fold, as in scaffold (**13**).

It seemed from the SAR studies (Figure 2) that the attachment of groups of a bulky nature to either side of both aryl parts ‘B’ and ‘C’ resulted in decreased inhibitory potentials against the *α*-glucosidase enzyme. When the –NO_2_ moiety of scaffold (**10**) present at the 4-position of aryl part ‘B’ was replaced by a bulkier –Br moiety, as in compound (**5**), the enzymatic activity against the *α*-glucosidase enzyme was lowered four-fold, showing that bulky natured substituents cause steric hindrance because of which the interaction caused by the synthetic analogs with the active residues of the *α*-glucosidase enzyme was diminished and, hence, the activity reduced. Among the scaffolds (**3**, **5**, **6,** and **17**), analog (**5**) bearing the –OH moiety at the 2-position and di-Cl moieties at the 3,5-position of aryl part ‘C’ along with a –Br group at the 4-position of aryl part ‘B’ was identified as the most potent inhibitor of *α*-glucosidase. Scaffold (**3**), which holds a –Br moiety at the 4-position of aryl part ‘B’ and a –NO_2_ group also at the 3-position of aryl part ‘C’, seemed to be a better competitor of *α*-glucosidase. However, its counterpart (**6**), which, although it holds substituents of the same nature, has a –NO_2_ group at the 4-position of aryl part ‘B’, demonstrated half the potency of compound (**3**) against the *α*-glucosidase enzyme. Moreover, the activity of scaffold (**5**) was decreased ten-fold by de-attachment of the –OH group followed by shifting of the 5-Cl to the 4-position of aryl part ‘C’, as in scaffold (**17**). This lowered activity of compound (**17**) in comparison to scaffold (**5**) may be attributed to the lack of a group (–OH) that can cause a strong interaction with the active amino acid residues through conventional hydrogen bonding. The scaffolds bearing the –Br moiety at aryl part ‘B’ and a –NO_2_ group either at the 4-position of aryl part ‘C’, as in compound (**6**), or at the 3-position of aryl part ‘C’ (**3**), displayed better potency against the *α*-glucosidase enzyme than compound (**17**) bearing di-Cl moieties at the 3,4-position of aryl part ‘C’ along with a –Br moiety at the 4-position of aryl part ‘B’. This superior potency of scaffolds (**6**) and (**3**) in comparison to scaffold (**17**) was due to better interaction of the –NO_2_ moiety through conventional hydrogen bonding with the active amino acid residues as well as the –NO_2_ group providing a strong e-withdrawing effect making the aryl part ‘C’ have a pi-cationic interaction with the active residues of *α*-glucosidase enzyme and, hence, enhanced the enzymatic activity. In addition, the attachment of substituents of stronger e-withdrawing effect (di-Cl) and a substituent (–OH) capable of interaction through stronger forces with the active residues of *α*-glucosidase, as in compound (**5**), enhanced the inhibitory potentials more greatly than compounds (**3**), (**6**) and (**17**) that lack substituents of this nature.

It was noteworthy that encouraging inhibitory potentials against the *α*-glucosidase enzymes were shown by scaffolds (**2**, **11**, **12,** and **14**) that hold a –NO_2_ moiety either at the 4-position or the 3-position of aryl part ‘C’ along with a variety of substitutions (–NO_2_, -OCH_3_ and di-Cl) present at different position of aryl part ‘B’ (Figure 3). Among these scaffolds, scaffold (**11**) proved to be a potent inhibitor of the *α*-glucosidase enzyme. This scaffold holds a –NO_2_ moiety on both aryl parts ‘B’ and ‘C’, but they have different positions on both sides of aryl parts ‘B’ and ‘C’. Aryl part ‘B’ has the –NO_2_ group at the 4-position while aryl part ‘C’ has the –NO_2_ moiety at the 3-position. The inhibition profile of scaffold (**11**) fell by half by shifting the –NO_2_ moiety of aryl part ‘C’ from the 3-position to the 4-position, as in compound (**12**), indicating that inhibitory potentials were affected by changing the position of the substituent around aryl part ‘C’. Moreover, it was also observed that the nature of the substituent also greatly affects the inhibitory activity of the synthetic scaffold; therefore, compound (**2**), which holds a –OCH_3_ moiety at the 3-position of aryl part ‘B’ along with a –NO_2_ moiety at the 3-position of aryl part ‘C’, displayed a two-fold lower potency than compound (**12**). Both compounds (**2**) and (**12**) hold the same –NO_2_ moiety at the 3-position of aryl part ‘C’ but have different substituents (–OCH_3_ and –NO_2_) around aryl part ‘B’. Furthermore, it was noted that the addition of di-Cl moieties at aryl part ‘B’ instead of the –OCH_3_ moiety resulted in enhanced activity, as in the case of compound (**14**), showing that the introduction of a substituent with a strong e-withdrawing nature in greater numbers makes the ring susceptible for interaction with the active residues of amino acid through pi-cationic interaction and, hence, enhanced the enzymatic activity (Figure 4).

Analog (**20**), which holds the –NO_2_ moiety at the 4-position of aryl ring ‘B’ and the –OH moiety also at the 4-position of aryl part ‘C’, emerged as a better competitor of *α*-glucosidase. This better potency shown by the –NO_2_ and –OH moieties-bearing analog was due to strong interaction with the active amino acid residues. An oxygen atom of both –OH and –NO_2_ has a strong tendency to bind well via strong conventional H-bonding with the inner cavity of *α*-glucosidase and hence, enhanced *α*-glucosidase activity. However, a decrease in the *α*-glucosidase activity of analog (**20**) was seemed by de-attachment of –OH moiety of aryl part ‘C’ and consequent attachment of di-Cl moieties at the 3,4-position of aryl part ‘C’ as in case of scaffold (**19**). This discrepancy in the inhibition profile of these scaffolds was because di-Cl moieties are incapable of interactions through a conventional hydrogen bond with the *α*-glucosidase active residues. Moreover, the activity was further decreased by attachment of a substituent of a bulky nature that is unable to interact better with the active part of the amino acid. Therefore, scaffold (**16**), which is different from analog (**20**) only in the nature of the substituent present at the 4-position of aryl part ‘B’, was found to be a low competitor of the *α*-glucosidase enzyme. This inferior potential of compound (**16**) was due to the attached bulky –Br moiety at the 4-position of aryl part ‘B’, even though this analog holds a –OH moiety at aryl part ‘C’. The bulky group causes steric hindrance and, hence, lowers the enzymatic inhibitory potentials. Furthermore, the inhibitory potential of scaffold (**18**) is better than that of analog (**16**) even though both these scaffolds hold the same substituents (–Br and –OH) around aryl parts ‘B’ and ‘C’ but have different positions of the –Br moiety at the aryl part ‘B’, indicating that the enzymatic activity was also dependent on the position of substituent around both aryl parts ‘B’ and ‘C’. Scaffold (**20**), bearing a –NO_2_ moiety at the 4-position of aryl part ‘B’ and a –OH moiety also at the 4-position of aryl part ‘C’, displayed better potency than scaffold (**18**), which holds a –OH moiety at the 4-position of aryl part ‘C’ but a –Br moiety at the 3-position of aryl part ‘B’ (Figure 5).

Overall, it was concluded that the inhibition profile of the benzotriazole-based bis-Schiff base was dependent mainly on the position and nature of substituents on both aryl parts ‘B’ and ‘C’. It was also observed that the alteration of the number/s of substituents around both aryl parts greatly affects the inhibitory potentials of the synthetic analogs.

### 2.3. Molecular Docking Study

The synthetic scaffolds and their measured inhibitions against α-glucosidase enzyme are listed in Table 1. It appeared from the IC_50_ values of benzotriazole-based bis-Schiff base scaffolds that α-glucosidase inhibition is related strongly to the position, nature, and number of attached functional moieties of both aryl parts ‘B’ and ‘C’ at the benzotriazole-based bis-Schiff base basic skeleton (Figure 1, Table 1). However, molecular docking was carried out to observe the position, nature, and number of attached substituents and enzymatic inhibition and further to develop the binding interactions of newly synthesized scaffolds with the active residues of the targeted α-glucosidase enzyme. It was observed from a detailed protein-ligand interaction (PLI) study of both potent scaffolds **38** and **10** against α-glucosidase that they established several key interactions with the active residues of the targeted enzyme, which may help in the improvement of the inhibition profile of these potent scaffolds against the targeted α-glucosidase enzyme. It was noteworthy that scaffold **9** adopted numerous interactions with the active amino acid residues of the α-glucosidase enzyme including Lys506 (pi-alkylation), Phe476 (pi-alkylation and pi-pi stacking), Asp232 (conventional hydrogen bond), Ala234 (conventional hydrogen bond and pi-alkylation), Asp568 (pi-anion), Arg552 (conventional hydrogen bond), ILE358 (pi-alkylation), Trp329 (pi-pi T shaped) and Trp432 (pi-pi stacking and pi-sigma) (Figure 6A).

Similarly, the PLI profile of the second most active scaffold **10** against the α-glucosidase enzyme also revealed numerous important interactions with the active sites of α-glucosidase including residues Ala234 (conventional hydrogen bond and pi-alkylation), Asp232 (conventional hydrogen bond), ILE233 (pi-alkylation), Phe236 (pi-alkylation and pi-pi- stacking), Phe476 (pi-pi stacking), Arg552 (conventional hydrogen bond), Asp568 (pi-anion), Trp329 (pi-pi stacking), Ph2601 (pi-pi T shaped), and Trp432 (pi-sigma) (Figure 7B). The high potentials of these active scaffolds might be due to attached –NO_2_ and di-Cl moieties. The –NO_2_ and di-Cl moieties on both sides of aryl parts ‘B’ and ‘C’ withdraw most of the electronic density from the aryl rings making them electron-deficient species, which further regain stability by adopting several key interactions with the active residues of the targeted enzyme and hence enhanced the enzymatic potential. Moreover, the attached –OH moiety also enhanced the enzymatic inhibition through participation in H-bonding with one of the active amino acid residues.

## 3. Materials and Methods

### 3.1. General Information

NMR spectra of benzotriazole derivatives that had been synthesized were acquired using a Bruker Ultra Shield FT NMR 600 MHz spectrometer operating at 600 MHz. HREI-MS was used to determine the mass, and fragmentation patterns were determined by analysis carried out using a Finnigan-MAT-311-A instrument. Thin layer chromatography was used to monitor the progress of reactions (Merck, Kieselgel 60F-254, 0.20 mm) and visualized using a UV lamp at 254 nm (UVGL58; Upland, CA, USA).

### 3.2. General Procedure for the Synthesis of Benzotriazole Scaffolds

#### 3.2.1. Formation of S-Substituted Benzotriazole Substrate (III)

In the first step, the pre-synthesized 1,2,3-benzotriazole (**I**) was treated with various phenacyl bromide (**II**) in ethanol and triethylamine, and the resulting mixture was refluxed with gentle stirring until the formation of the 2-(1H-benzo[*d*][1,2,3]triazol-1-yl)-1-phenylethan-1-one substrate (**III**) was complete (the conversion was monitored by TLC, reflux, 3 h). After completion of the reaction, the residue was stored overnight on cooling for crystallization of the desired substrate (**III**)**.**

#### 3.2.2. Synthesis of Benzotriazole-Based Bis-Schiff Base Scaffolds (**1**–**20**)

In the next step, the carbonyl residue of substrate (**III**) underwent condensation with hydrated hydrazine in acetic acid, and the resulting mixture was stirred for 3 hrs to achieve the formation of (E)-1-(2-hydrazono-2-phenylethyl)-1H-benzo[*d*][1,2,3]triazole (**IV**)**,** which was treated with various benzaldehydes in acetic acid to afford the synthesis of 1,2,3-benzotriazole based bis-Schiff base derivatives (**1**–**20**)**.** The synthesized scaffolds were characterized by ^1^H NMR, ^13^C NMR, and HREI-MS spectroscopy.

### 3.3. Spectral Analysis

#### 3.3.1. 2-((E)-(((E)-2-(1H-benzo[d][1,2,3]triazol-1-yl)-1-(3-methoxyphenyl)ethylidene) hydrazono)methyl)-4,6-dichlorophenol (**1**)

Yield: 69%; yellow solid; m.p.: 195–196 °C; ^1^H-NMR (600 MHz,DMSO-*d_6_*): *δ* 9.56 (s, 1H, OH), 7.98 (dd, *J* = 7.2, 1.9 Hz, 1H, benzotriazole-H), 7.93 (dd, *J* = 7.4, 1.6 Hz, 1H, benzotriazole-H), 7.62 (dt, *J* = 6.8, 1.8 Hz, 1H, aromatic-H), 7.40 (m, 2H, benzotriazole-H), 7.51 (d, *J* = 1.6 Hz, 1H, aromatic-H), 7.48 (d, *J* = 2.2 Hz, 1H, aromatic-H), 7.42 (t, *J* = 1.9 Hz, 1H, aromatic-H), 7.32 (t, *J* = 2.1 Hz, 1H, aromatic-H), 7.07 (dt, *J* = 6.9 Hz, 1H, aromatic-H), 6.97 (s, 1H, N=CH), 3.73 (s, 3H, CH_3_), 3.40 (s, 2H, CH_2_); ^13^C-NMR (125 MHz, DMSO-*d_6_*): *δ* 172.1 (C=N), 167.0 (C-OH), 165.0 (C-O), 155.8 (C=N), 152.2 (C-Cl), 148.4 (C-Cl), 138.3 (C-Cl), 134.2 (C), 133.0 (C), 132.4 (C-N), 128.3 (C-Cl), 125.1 (C-Cl), 123.6 (CH), 122.4 (CH), 121.5 (CH), 119.4 (CH), 118.7 (CH), 118.4 (CH), 117.1 (CH), 116.5 (C), 55.3 (CH_3_-O), 50.7 (CH_2_);HREIMS: *m*/*z* calcd for C_22_H_17_Cl_2_N_5_O_2_ [M]^+^ 453.0759; Found; 453.0630.

#### 3.3.2. 1-((E)-2-(3-methoxyphenyl)-2-(((E)-3-nitrobenzylidene)hydrazono)ethyl)-1H-benzo[d] [1,2,3]triazole (**2**)

Yield: 75%; yellow solid; m.p.: 192–193 °C; ^1^H-NMR (600 MHz, DMSO-*d_6_*): *δ* 8.24-8.16 (m, 1H, aromatic-H), 8.15 (t, *J* = 8.8 Hz, 1H, aromatic-H), 7.95 (dd, *J* = 8.2, 3.0 Hz, 1H, aromatic-H), 7.84–7.78 (m, 1H, aromatic-H), 7.75 (d, *J* = 8.1 Hz, 2H, benzotriazole-H), 7.72 (s, 1H, aromatic-H), 7.70 (dd, *J* = 7.0, 2.3 Hz, 1H, aromatic-H), 6.72 (s, 1H, aromatic-H), 6.61 (s, 1H, N=CH), 6.59 (d, *J* = 6.3 Hz, 2H, benzotriazole-H), 2.11 (s, 2H, CH_2_), 1.55 (s, 3H, CH_3_); ^13^C-NMR (150 MHz, DMSO-*d_6_*): *δ* 169.8 (C=N), 156.1 (C-OCH_3_), 145.8 (C=N), 136.1 (C-NO_2_), 126.8 (C-N), 125.5 (C), 125.5 (C), 125.1 (CH), 125.4 (C-N), 125.0 (CH), 124.8 (CH), 124.6 (CH), 124.0 (CH), 121.6 (CH), 119.0 (CH), 119.0 (CH), 118.3 (CH), 118.1 (CH), 118.0 (CH), 112.8 (C), 46.2 (CH_2_), 41.2 (O-CH_3_).; HREI-MS: *m*/*z* calcd for C_22_H_18_N_6_O_3_[M]^+^ 414.1436; Found; 414.1305.

#### 3.3.3. 1-((E)-2-(4-bromophenyl)-2-(((E)-3-nitrobenzylidene)hydrazono)ethyl)-1H-benzo[d] [1,2,3]triazole (**3**)

Yield: 68%; black solid; m.p.: 196–197 °C; ^1^H-NMR (600 MHz, DMSO-*d_6_*): *δ* 8.46 (s, 1H, aromatic-H ), 8.17 (dt, *J* = 8.1, 2.2 Hz, 1H, aromatic-H), 8.12 (dd, *J* = 7.9, 1.8 Hz, 1H, aromatic-H), 7.96 (dd, *J* = 7.8, 2.1 Hz, 1H, benzotriazole-H ), 7.89 (dd, *J* = 8.1, 1.9 Hz, 1H, benzotriazole-H ), 7.74 (t, *J* = 7.5 Hz, 1H, aromatic-H), 7.68 (d, *J* = 7.6 Hz, 2H, aromatic-H), 7.60 (d, *J* = 6.9 Hz, 2H, aromatic-H), 7.42 (m, 2H, benzotriazole-H), 7.17 (s, 1H, N=CH ), 3.78 (s, 2H, CH_2_); ^13^C-NMR (150 MHz, DMSO-*d_6_*): *δ* 180.0 (C=N), 164.3 (C=N), 153.1 (C-NO_2_), 144.2 (C-Br), 142.8 (C-N), 136.8 (C), 135.6 (C), 134.1 (CH), 130.0 (C-N), 129.8 (CH), 129.7 (CH), 126.1 (CH), 125.8 (CH), 124.4 (CH), 123.3 (CH), 123.1 (CH), 122.8 (CH), 122.2 (CH), 121.1 (CH), 121.0 (C), 49.2 (CH_2_).;HREI-MS: *m*/*z* calcd for C_21_H_15_BrN_6_O_2_ [M]^+^ 462.0436; Found; 462.0213.

#### 3.3.4. 1-((E)-2-(((E)-anthracen-9-ylmethylene)hydrazono)-2-(4-bromophenyl)ethyl)-1H-benzo[d] [1,2,3]triazole (**4**)

Yield: 65%; red solid; m.p.: 201–202 °C; ^1^H-NMR (600 MHz, DMSO-*d_6_*): *δ* 8.70 (s, 1H, aromatic-H), 8.03 (dd, *J* =7.9, 2.4 Hz, 2H, aromatic-H), 7.98 (dd, *J* = 7.7, 2.1 Hz, 1H, benzotriazole-H), 7.84 (dd, *J* = 7.7, 1.8 Hz, 2H, aromatic-H), 7.72 (d, *J* = 7.5 Hz, 2H, aromatic-H), 7.66 (d, *J* = 7.6 Hz, 2H, aromatic-H), 7.48 (m, 4H, aromatic-H), 7.42 (m, 2H, benzotriazole-H), 7.20 (s, 1H, N=CH), 3.38 (s, 2H, CH_2_); ^13^C-NMR (150 MHz, DMSO-*d_6_*): *δ* 173.1 (C=N), 171.2 (C=N), 165.1 (C-Br), 163.2 (C), 155.8 (C-N), 153.3 (C), 152.8 (C), 149.6 (C), 148.1 (C), 147.0 (C-N), 146.4 (C), 142.8 (C), 141.7 (C), 140.1 (CH), 138.8 (CH), 137.4 (CH), 136.3 (CH), 133.1 (CH), 132.8 (CH), 131.2 (CH), 130.1 (CH), 129.5 (CH), 126.3 (CH), 124.0 (CH), 125.1 (CH), 123.8 (CH), 122.2 (CH),120.3 (CH), 46.6 (CH_2_).;HREI-MS: *m*/*z* calcd for C_29_H_20_BrN_5_ [M]^+^ 517.0902; Found; 517.0807.

#### 3.3.5. 2-((E)-(((E)-2-(1H-benzo[d][1,2,3]triazol-1-yl)-1-(4-bromophenyl)ethylidene) hydrazono) methyl)-4,6-dichlorophenol (**5**)

Yield: 63%; white solid; m.p.: 196–197 °C; ^1^H-NMR (600 MHz, DMSO-*d_6_*): *δ* 9.58 (s, 1H, OH), 7.96 (dd, *J* = 7.7, 2.2 Hz, 1H, benzotriazole-H), 7.94 (dd, *J* = 7.8, 2.1 Hz, 1H, benzotriazole-H), 7.70 (d, *J* = 7.2 Hz, 2H, aromatic-H), 7.60 (d, *J* = 6.8 Hz, 2H, aromatic-H), 7.54 (s, 1H, aromatic-H), 7.48 (s, 1H, aromatic-H), 7.43 (m, 2H, benzotriazole-H), 7.14 (s, 1H, N=CH), 3.42 (s, 2H, CH_2_); ^13^C-NMR (150 MHz, DMSO-*d_6_*): *δ* 174.1 (C=N), 168.0 (C-OH), 166.0 (C-Br), 156.8 (C=N), 155.2 (C-Cl), 152.4 (C-Cl), 147.3 (CH), 145.2 (C), 143.0 (C), 141.4 (C-N), 140.3 (CH), 139.1 (CH), 136.6 (CH), 133.4 (CH), 127.5 (CH), 124.4 (CH), 121.7 (CH), 120.4 (CH), 119.1 (CH), 118.5 (C), 50.3 (CH_2_).; HREI-MS: *m*/*z* calcd for C_21_H_14_BrCl_2_N_5_O [M]^+^ 500.9759; Found; 500.9584.

#### 3.3.6. 1-((E)-2-(4-bromophenyl)-2-(((E)-4-nitrobenzylidene)hydrazono)ethyl)-1H-benzo[d] [1,2,3]triazole (**6**)

Yield: 78%; brown solid; m.p.: 196-197 °C; ^1^H-NMR (600 MHz, DMSO-*d_6_*): *δ* 8.35 (d, *J* = 7.9 Hz, 2H, aromatic-H), 8.23 (d, *J* = 6.4 Hz, 2H, aromatic-H), 7.95 (d, *J* = 6.6 Hz, 1H, aromatic-H), 7.85 (d, *J* = 8.1 Hz, 1H, aromatic-H), 7.73 (d, *J* = 7.8 Hz, 2H, benzotriazole-H), 6.75 (s, 1H, N=CH), 6.62 (d, *J* = 6.5 Hz, 2H, benzotriazole-H), 2.67 (s, 2H, CH_2_), ^13^C-NMR (150 MHz, DMSO-*d_6_*): *δ* 180.1 (C=N), 166.2 (C=N), 155.1 (C-NO_2_), 146.2 (C-Br), 145.8 (C-N), 139.8 (C), 139.6 (C), 136.1 (CH), 131.0 (C-N), 130.8 (CH), 130.7 (CH), 127.1 (CH), 126.8 (CH), 125.4 (CH), 125.3 (CH), 125.1 (CH), 124.8 (CH), 124.2 (CH), 124.1 (CH), 122.5 (C), 46.2 (CH_2_).; HREI-MS: *m*/*z* calcd for C_21_H_15_BrN_6_O_2_S [M]^+^ 462.0438; Found; 462.0216.

#### 3.3.7. 1-((E)-2-(4-bromophenyl)-2-(((E)-naphthalen-2-ylmethylene)hydrazono)ethyl)-1H-benzo[d] [1,2,3]triazole (**7**)

Yield: 71%; light yellow solid; m.p.: 196–197 °C; ^1^H-NMR (600 MHz, DMSO-*d_6_*): *δ* 8.58 (s, 1H, aromatic-H), 8.49 (dd, *J* = 8.2, 2.2 Hz, 1H, aromatic-H), 8.42 (s, 1H, N = CH), 8.01 (dd, *J* = 8.1, 2.3 Hz, 1H, aromatic-H), 8.00 (dd, *J* = 7.9, 2.3 Hz, 1H, aromatic-H), 7.96 (d, *J* = 6.9 Hz, 1H, aromatic-H), 7.95 (dd, J = 7.8, 2.1 Hz, 1H, benzotriazole-H), 7.89 (dd, *J* = 7.7, 1.9 Hz, 1H, benzotriazole-H), 7.72 (d, *J* = 7.5 Hz, 2H, aromatic-H), 7.66 (d, *J* = 7.4 Hz, 2H, aromatic-H), 7.57 (m, 2H, aromatic-H), 7.42 (m, 2H, benzotriazole-H), 3.40 (s, 2H, CH_2_); ^13^C-NMR (150 MHz, DMSO-*d_6_*): *δ* 173.1 (C=N), 171.2 (C=N), 165.1 (C-Br), 163.2 (C), 155.8 (C-N), 153.3 (C), 152.8 (C), 149.6 (C), 148.1 (C), 147.0 (C-N), 142.8 (C), 141.7 (C), 140.1 (CH), 138.8 (CH), 137.4 (CH), 136.3 (CH), 133.1 (CH), 132.8 (CH), 131.2 (CH), 130.1 (CH), 129.5 (C), 125.1 (CH), 123.8 (CH), 122.2 (CH), 46.6 (CH_2_).; HREI-MS: *m*/*z* calcd for C_25_H_18_BrN_5_ [M]^+^ 467.0746; Found; 467.0630.

#### 3.3.8. 4-((E)-(((E)-2-(1H-benzo[d][1,2,3]triazol-1-yl)-1-(4-bromophenyl)ethylidene) hydrazono) methyl)benzaldehyde (**8**)

Yield: 61%; light red solid; m.p.: 193–194 °C; ^1^H-NMR (600 MHz, DMSO-*d_6_*): *δ* 9.83 (s, 1H, CH), 8.43 (s, 1H, N = CH), 7.96 (dd, *J* = 7.8, 2.1 Hz, 1H, benzotriazole-H), 8.01 (d, *J* = 7.9 Hz, 1H, aromatic-H), 7.87 (dd, *J* = 7.6, 1.9 Hz, 1H, benzotriazole-H), 7.84 (d, *J* = 7.7 Hz, 2H, aromatic-H), 7.72 (d, *J* = 7.3Hz, 2H, aromatic-H), 7.66 (d, *J* = 7.6 Hz, 2H, aromatic-H), 7.41 (m, 2H, benzotriazole-H), 7.38 (d, *J* = 7.2 Hz, 2H, aromatic-H), 6.94 (s, 1H, N=CH), 4.62 (s, 2H, CH_2_); ^13^C-NMR (150 MHz, DMSO-*d_6_*): *δ* 175.1 (C=N), 170.2 (C=N), 168.1 (C=O), 162.2 (C-Br), 158.8 (C-N), 154.3 (C), 153.8 (C), 144.6 (C), 143.1 (CH), 141.0 (C-N), 140.8 (CH), 136.7 (CH), 137.1 (CH), 136.8 (CH), 129.4 (CH), 126.3 (CH), 124.1 (CH), 123.8 (CH), 122.2 (CH), 121.1 (CH), 120.5 (C), 46.9 (CH_2_).; HREI-MS: *m*/*z* calcd for C_22_H_16_BrN_5_O [M]^+^ 445.0538; Found; 445.0364.

#### 3.3.9. 2-((E)-(((E)-2-(1H-benzo[d][1,2,3]triazol-1-yl)-1-(3-nitrophenyl)ethylidene) hydrazono)methyl)-4,6-dichlorophenol (**9**)

Yield: 73%; purple solid; m.p.: 197–198 °C; ^1^H-NMR (600 MHz, DMSO-*d_6_*): *δ* 9.54 (s, 1H, OH), 8.54 (s, 1H, aromatic-H), 8.42 (s, 1H, N=CH), 8.35 (dd, *J* = 7.8, 2.2 Hz, 1H, aromatic-H), 8.17 (dd, *J* = 7.6, 1.9 Hz, 1H, aromatic-H), 7.95 (dd, *J* = 7.8, 2.2 Hz, 1H, benzotriazole-H), 7.87 (dd, J = 7.7, 1.8 Hz, 1H, benzotriazole-H), 7.66 (t, *J* = 7.5 Hz, 1H, aromatic-H), 7.52 (d, *J* = 1.9 Hz, 1H, aromatic-H), 7.46 (d, *J* = 2.1 Hz, 1H, aromatic-H), 7.42 (m, 2H, benzotriazole-H), 3.40 (s, 2H, CH_2_); ^13^C-NMR (150 MHz, DMSO-*d_6_*): *δ* 174.4 (C=N), 168.5 (C-OH), 166.4 (C-NO_2_), 156.3 (C=N), 155.9 (C-Cl), 152.7 (C-Cl), 147.8 (CH), 145.0 (C), 143.3 (C), 141.1 (C-N), 140.8 (CH), 139.4 (CH), 136.3 (CH), 133.1 (CH), 127.4 (CH), 124.2 (CH), 121.5 (CH), 120.3 (CH), 119.0 (CH), 118.7 (C), 50.6 (CH_2_).; HREI-MS: *m*/*z* calcd for C_21_H_14_Cl_2_N_6_O_3_ [M]^+^ 468.0504; Found; 468.0385.

#### 3.3.10. 2-((E)-(((E)-2-(1H-benzo[d][1,2,3]triazol-1-yl)-1-(4-nitrophenyl)ethylidene) hydrazono)methyl)-4,6-dichlorophenol (**10**)

Yield: 70%; black solid; m.p.: 197–198 °C; ^1^H-NMR (600 MHz, DMSO-*d_6_*): *δ* 9.54 (s, 1H, OH), 8.33 (d, *J* = 7.8 Hz, 2H, aromatic-H), 8.05 (d, *J* = 7.6 Hz, 2H, aromatic-H), 7.95 (dd, *J* = 7.8, 2.2 Hz, 1H, benzotriazole-H), 7.87 (dd, *J* = 7.7, 1.8 Hz, 1H, benzotriazole-H), 7.51 (d, *J* = 1.9 Hz, 1H, aromatic-H), 7.45 (d, *J* = 2.1 Hz, 1H, aromatic-H), 7.42 (m, 2H, benzotriazole-H), 7.12 (s, 1H, N=CH), 3.41 (s, 2H, CH_2_); ^13^C-NMR (150 MHz, DMSO-*d_6_*): *δ* 175.4 (C=N), 167.5 (C-OH), 165.4 (C-NO_2_), 156.2 (C=N), 155.8 (C-Cl), 152.6 (C-Cl), 147.7 (CH), 145.5 (C), 143.0 (C), 141.3 (C-N), 140.6 (CH), 139.3 (CH), 136.2 (CH), 133.0 (CH), 127.2 (CH), 124.5 (CH), 121.4 (CH), 120.1 (CH), 119.4 (CH), 118.8 (C), 50.3 (CH_2_).;HREI-MS: *m*/*z* calcd for C_21_H_14_Cl_2_N_6_O_3_ [M]^+^ 468.0504; Found; 468.0385.

#### 3.3.11. 1-((E)-2-(((E)-3-nitrobenzylidene)hydrazono)-2-(4-nitrophenyl)ethyl)-1H-benzo[d] [1,2,3]triazole (**11**)

Yield: 72%; white solid; m.p.: 191–192 °C; ^1^H-NMR (600 MHz, DMSO-*d_6_*): *δ* 8.44 (d, *J* =1.8 Hz,1H, aromatic-H), 8.40 (s, 1H, N=CH), 8.36 (d, *J* = 7.9 Hz, 2H, aromatic-H), 8.16 (m, 1H, aromatic-H), 8.07 (dd, *J* = 7.9,1.8 Hz, 1H, aromatic-H), 8.04 (d, *J* = 7.6 Hz, 2H, aromatic-H), 7.96 (dd, *J* = 7.3, 2.2 Hz, 1H, benzotriazole-H), 7.86 (dd, *J* = 7.9, 1.8 Hz, 1H, benzotriazole-H), 7.72 (t, *J* = 7.6 Hz, 1H, aromatic-H),7.42 (m, 2H, benzotriazole-H), 3.40 (s, 2H, CH_2_); ^13^C-NMR (150 MHz, DMSO-*d_6_*): *δ* 169.6 (C=N), 156.8 (C-NO_2_), 145.5 (C=N), 136.4 (C-NO_2_), 126.4 (C-N), 125.3 (C), 125.1 (C), 124.7 (CH), 124.4 (C-N), 124.0 (CH), 123.3 (CH), 123.2 (CH), 122.0 (CH), 121.1 (CH), 119.5 (CH), 119.4 (CH), 118.7 (CH), 118.4 (CH), 118.1 (CH), 112.5 (C), 46.7 (CH_2_).; HREI-MS: *m*/*z* calcd for C_21_H_15_N_7_O_4_ [M]^+^ 429.1186; Found; 429.1026.

#### 3.3.12. 1-((E)-2-(((E)-4-nitrobenzylidene)hydrazono)-2-(4-nitrophenyl)ethyl)-1H-benzo[d] [1,2,3]triazole (**12**)

Yield: 81%; brown solid; m.p.: 191–192 °C; ^1^H-NMR (600 MHz, DMSO-*d_6_*): *δ* 8.36 (d, *J* = 8.2, 2.1 Hz, 2H, aromatic-H), 8.30 (d, *J* = 8.1 Hz, 2H, aromatic-H), 8.15 (d, *J* = 7.9 Hz, 2H, aromatic-H), 8.09 (d, *J* = 7.8 Hz, 2H, aromatic-H), 7.96 (dd, *J* = 7.8, 1.7 Hz, 1H, benzotriazole-H), 7.87 (dd, *J* = 7.9, 1.8 Hz, 1H, benzotriazole-H), 7.42 (m, 2H, benzotriazole-H), 6.99 (s, 1H, N=CH), 3.40 (s, 2H, CH_2_); ^13^C-NMR (150 MHz, DMSO-*d_6_*): *δ* 178.1 (C=N), 165.2 (C=N), 154.1 (C-NO_2_), 147.2 (C-NO_2_), 143.8 (C-N), 138.8 (C), 137.6 (C), 135.1 (CH), 130.0 (C-N), 129.8 (CH), 129.8 (CH), 125.8 (CH), 125.8 (CH), 124.4 (CH), 123.3 (CH), 122.1 (CH), 122.1 (CH), 121.2 (CH), 121.2 (CH), 120.5 (C), 48.2 (CH_2_).; HREI-MS: *m*/*z* calcd for C_21_H_15_N_7_O_4_ [M]^+^ 429.1186; Found; 429.1026.

#### 3.3.13. 2-((E)-(((E)-2-(1H-benzo[d][1,2,3]triazol-1-yl)-1-(2,4-dichlorophenyl)ethylidene) hydrazono)methyl)-4,6-dichlorophenol (**13**)

Yield: 76%; yellow solid; m.p.: 199–200 °C; ^1^H-NMR (600 MHz, DMSO-*d_6_*): *δ* 9.97 (s, 1H, OH), 7.59 (d, *J* = 8.0 Hz, 1H, aromatic-H), 7.52 (d, *J* = 7.9 Hz, 1H, aromatic-H), 7.47 (d, *J* = 1.9 Hz, 1H, aromatic-H), 7.31 (s, 1H, aromatic-H), 7.30 (d, *J* = 6.8 Hz, 2H, benzotriazole-H), 7.18 (s, 1H, aromatic-H), 7.02 (d, *J* = 7.4 Hz, 2H, benzotriazole-H), 7.00 (s, 1H, N=CH), 3.64 (s, 2H, CH_2_); ^13^C-NMR (150 MHz, DMSO-*d_6_*): *δ* 180.0 (C=N), 168.8 (C=N), 140.2 (C-OH), 139.9 (C-Cl), 139.7 (C-Cl), 129.8 (C), 129.2 (C), 129.1 (C-Cl), 128.8 (C-Cl), 124.8 (C-N), 124.0 (CH), 124.0 (CH), 123.4 (CH), 123.2 (CH), 123.0 (CH), 122.5 (CH), 122.2 (CH), 119.4 (CH), 118.6 (CH), 118.5 (C), 46.1 (CH_2_).; HREI-MS: *m*/*z* calcd for C_21_H_13_Cl_4_N_5_O [M]^+^ 492.9845; Found; 492.9691.

#### 3.3.14. 1-((E)-2-(2,4-dichlorophenyl)-2-(((E)-3-nitrobenzylidene)hydrazono)ethyl)-1H-benzo[d] [1,2,3]triazole (**14**)

Yield: 74%; dark yellow solid; m.p.: 194–195 °C; ^1^H-NMR (600 MHz, DMSO-*d_6_*): *δ* 8.42 (d, *J* = 1.8 Hz, 1H, aromatic-H), 8.36 (s, 1H, N=CH), 8.12 (dd, *J* = 7.7, 1.9 Hz, 1H, aromatic-H), 8.05 (dd, *J* = 8.1, 1.9 Hz, 1H, aromatic-H), 7.94 (dd, *J* = 8.2, 2.3 Hz, 1H, benzotriazole-H), 7.86 (dd, *J* = 8.5, 2.3 Hz, 1H, benzotriazole-H), 7.70 (d, *J* = 8.2 Hz, 1H, aromatic-H), 7.68 (t, *J* = 7.6Hz, 1H, aromatic-H), 7.66 (s, 1H, aromatic-H), 7.34 (d, *J* = 7.1 Hz, 1H, aromatic-H), 7.36 (m, 2H, benzotriazole-H), 3.34 (s, 2H, CH_2_); ^13^C-NMR (150 MHz, DMSO-*d_6_*): *δ* 169.1 (C=N), 156.0 (C-NO_2_), 145.8 (C=N), 136.2 (C-NO_2_), 126.4 (C-NO_2_), 124.3 (C-Cl), 125.2 (C), 125.0 (C), 124.4 (C-N), 124.3 (C-Cl), 123.1 (CH), 123.0 (CH), 121.6 (CH), 121.4 (CH), 120.5 (CH), 120.4 (CH), 117.7 (CH), 117.4 (CH), 116.1 (CH), 115.5 (C), 47.7 (CH_2_).; HREI-MS: *m*/*z* calcd for C_21_H_14_Cl_2_N_6_O_2_ [M]^+^ 452.0555; Found; 452.0401.

#### 3.3.15. 1-((E)-2-(2,4-dichlorophenyl)-2-(((E)-4-nitrobenzylidene)hydrazono)ethyl)-1H-benzo[d] [1,2,3]triazole (**15**)

Yield: 64%; red solid; m.p.: 194–195 °C; ^1^H-NMR (600 MHz, DMSO-*d_6_*): *δ* 8.35 (d, *J* = 8.1 Hz, 2H, aromatic-H), 8.10 (d, *J* = 7.8 Hz, 2H, aromatic-H), 7.92 (dd, *J* = 7.7, 2.1 Hz, 1H, benzotriazole-H), 7.88 (dd, *J* = 7.6, 2.3 Hz, 1H, benzotriazole-H), 7.73 (s, 1H, aromatic-H), 7.64 (d, *J* = 7.4 Hz, 1H, aromatic-H), 7.39 (m, 2H, benzotriazole-H), 7.32 (d, *J* = 7.2 Hz, 1H, aromatic-H), 6.96 (s, 1H, N=CH), 3.87 (s, 2H, CH_2_); ^13^C-NMR (150 MHz, DMSO-*d_6_*): *δ* 169.7 (C=N), 168.4 (CH), 167.3 (C-NO_2_), 166.0 (C=N), 164.7 (C-Cl), 163.5 (C-Cl), 160.6 (CH), 155.4 (C), 153.5 (C), 151.2 (C-N), 150.5 (CH), 146.2 (CH), 142.1 (CH), 138.0 (CH), 134.6 (CH), 127.3 (CH), 126.2 (CH), 124.0 (CH), 120.3 (CH), 119.7 (C), 49.6 (CH_2_).; HREI-MS: *m*/*z* calcd for C_21_H_14_Cl_2_N_6_O_2_ [M]^+^ 452.0555; Found; 452.0401.

#### 3.3.16. 4-((E)-(((E)-2-(1H-benzo[d][1,2,3]triazol-1-yl)-1-(4-bromophenyl)ethylidene) hydrazono)methyl)benzaldehyde (**16**)

Yield: 71%; white solid; m.p.: 193–194 °C; ^1^H-NMR (600 MHz, DMSO-*d_6_*): *δ* 9.67 (s, 1H, CH), 8.25 (d, *J* = 8.4 Hz, 2H, aromatic-H), 7.74 (d, *J* = 7.5 Hz, 2H, aromatic-H), 7.68 (d, *J* = 7.2 Hz, 2H, aromatic-H), 7.43 (d, *J* = 8.3 Hz, 2H, aromatic-H), 7.24 (d, *J* = 7.3 Hz, 2H, benzotriazole-H), 7.20 (d, *J* = 7.6 Hz, 2H, benzotriazole-H), 7.17 (s, 1H, N=CH), 3.37 (s, 2H, CH_2_); ^13^C-NMR (150 MHz, DMSO-*d_6_*): *δ*181.8 (C=N), 166.2 (C=N), 138.8 (C-OH), 133.0 (C-Br), 132.5 (C-N), 128.7 (C), 128.5 (C), 128.2 (CH), 126.8 (C-N), 124.8 (CH), 123.7 (CH), 123.1 (CH), 122.8 (CH), 122.4 (CH), 122.3 (CH), 122.1 (CH), 121.9 (CH), 121.4 (CH), 121.3 (CH), 112.9 (C), 46.1 (CH_2_).; HREI-MS: *m*/*z* calcd for C_22_H_16_BrN_5_O [M]^+^ 445.0538; Found; 445.0334.

#### 3.3.17. 1-((E)-2-(4-bromophenyl)-2-(((E)-3,4-dichlorobenzylidene)hydrazono)ethyl)-1H-benzo[d] [1,2,3]triazole (**17**)

Yield: 63%; black solid; m.p.: 198–199 °C; ^1^H-NMR (600 MHz, DMSO-*d_6_*): *δ* 8.28 (s, 1H, aromatic-H), 7.96 (dd, *J* = 7.8, 2.4 Hz,1H, benzotriazole-H), 7.87 (dd, *J* = 8.1, 2.3 Hz, 1H, benzotriazole-H), 7.84 (d, *J* = 7.2 Hz, 1H, aromatic-H),7.74 (d, *J* = 6.8 Hz, 2H, aromatic-H), 7.62 (d, *J* = 6.6 Hz, 2H, aromatic-H), 7.61 (d, *J* = 7.5 Hz, 1H, aromatic-H), 7.40 (m, 2H, benzotriazole-H), 6.93 (s, 1H, N=CH), 3.85 (s, 2H, CH_2_); ^13^C-NMR (150 MHz, DMSO-*d_6_*): *δ* 169.2 (C=N), 168.1 (CH), 167.4 (C-Br), 166.5 (C=N), 164.7 (C-Cl), 163.4 (C-Cl), 160.3 (CH), 155.0 (C), 153.3 (C), 151.7 (C-N), 150.5 (CH), 146.2 (CH), 142.1 (CH), 139.8 (CH), 134.3 (CH), 127.4 (CH), 126.2 (CH), 124.0 (CH), 120.3 (CH), 119.7 (C), 48.4 (CH_2_).; HREI-MS: *m*/*z* calcd for C_21_H_14_BrC_2_N_5_ [M]^+^ 484.9810; Found; 484.9627.

#### 3.3.18. 4-((E)-(((E)-2-(1H-benzo[d][1,2,3]triazol-1-yl)-1-(3-bromophenyl)ethylidene) hydrazono) methyl)benzaldehyde (**18**)

Yield: 61%; brown solid; m.p.: 193–194 °C; ^1^H-NMR (600 MHz, DMSO-*d_6_*): *δ* 9.82 (s, 1H, OH), 8.00 (d, *J* = 7.6 Hz, 2H, aromatic-H), 7.97 (dd, *J* = 7.9, 2.0 Hz, 1H, benzotriazole-H), 7.89 (s, 1H, aromatic-H), 7.86 (dd, *J* = 7.9, 1.8 Hz, 1H, benzotriazole-H), 7.85 (d, *J* = 7.5 Hz, 2H, aromatic-H), 7.71 (dd, *J* = 8.1, 1.8 Hz, 1H, aromatic-H), 7.59 (dd, *J* = 7.6, 2.3 Hz, 2H, aromatic-H), 7.42 (m, 2H, benzotriazole-H), 7.39 (t, 1H, aromatic-H), 6.42 (s, 1H, N=CH), 4.64 (s, 2H, CH_2_); ^13^C-NMR (150 MHz, DMSO-*d_6_*): *δ*181.3 (C=N), 166.0 (C=N), 138.5 (C-OH), 133.3 (C-Br), 132.4 (C-N), 128.6 (C), 128.3 (C), 128.1 (CH), 126.7 (C-N), 124.6 (CH), 123.5 (CH), 123.2 (CH), 122.6 (CH), 122.3 (CH), 122.0 (CH), 121.7 (CH), 121.4 (CH), 121.0 (CH), 120.3 (CH), 112.8 (C), 47.4 (CH_2_).; HREI-MS: *m*/*z* calcd for C_22_H_16_BrN_5_O [M]^+^ 445.0538; Found; 445.0334.

#### 3.3.19. 1-((E)-2-(((E)-3,4-dichlorobenzylidene)hydrazono)-2-(4-nitrophenyl)ethyl)-1H-benzo[d] [1,2,3]triazole (**19**)

Yield: 79%; black solid; m.p.: 194–195 °C;^1^H-NMR (600 MHz, DMSO*-d_6_*): *δ* 8.29 (s, 1H, aromatic-H), 7.98 (dd, *J* = 7.9, 2.3 Hz,1H, benzotriazole-H), 7.88 (dd, *J* = 8.2, 2.2 Hz, 1H, benzotriazole-H), 7.85 (d, *J* = 7.4 Hz, 1H, aromatic-H), 7.75 (d, *J* = 6.9 Hz, 2H, aromatic-H), 7.64 (d, *J* =6.7 Hz, 2H, aromatic-H), 7.63 (d, *J* = 7.6 Hz, 1H, aromatic-H), 7.41 (m, 2H, benzotriazole-H), 7.10 (s, 1H, N=CH), 3.89 (s, 2H, CH_2_); ^13^C-NMR (150 MHz, DMSO-*d_6_*): *δ* 169.4 (C=N), 168.5 (CH), 167.4 (C-NO_2_), 166.2 (C=N), 164.8 (C-Cl), 163.6 (C-Cl), 160.7 (CH), 155.5 (C), 153.0 (C), 151.3 (C-N), 150.6 (CH), 146.3 (CH), 142.2 (CH), 139.0 (CH), 134.2 (CH), 127.5 (CH), 126.4 (CH), 124.1 (CH), 120.4 (CH), 119.8 (C), 49.7 (CH_2_).; HREI-MS: *m*/*z* calcd for C_21_H_14_Cl_2_N_6_O_2_ [M]^+^ 452.0555; Found; 452.0401.

#### 3.3.20. 4-((E)-(((E)-2-(1H-benzo[d][1,2,3]triazol-1-yl)-1-(4-nitrophenyl)ethylidene) hydrazono)methyl)phenol (**20**)

Yield: 80%; brown solid; m.p.: 190-191 °C; ^1^H-NMR (600 MHz, DMSO-*d_6_*): *δ* 9.90 (s, 1H, OH), 8.26 (d, *J* = 7.0 Hz, 2H, aromatic-H), 7.96 (d, *J* = 8.4 Hz, 2H, aromatic-H), 7.59 (d, *J* = 7.6 Hz, 2H, aromatic-H), 7.42 (d, *J* = 6.8 Hz, 2H, aromatic-H), 7.00 (s, 1H, N=CH), 6.74 (d, *J* = 6.6 Hz, 2H, benzotriazole-H), 6.60 (d, J = 7.8 Hz, 2H, benzotriazole-H), 3.37 (s, 2H, CH_2_); ^13^C-NMR (150 MHz, DMSO-*d_6_*): *δ* 180.1 (C=N), 169.4 (C=N), 149.5 (C-OH), 140.0 (C-Br), 139.7 (C-N), 139.7 (C), 131.5 (C), 131.3 (CH), 131.0 (C-N), 130.9 (CH), 130.7 (CH), 126.9 (CH), 125.5 (CH), 124.9 (CH), 124.2 (CH), 124.0 (CH), 120.0 (CH), 120.0 (CH), 119.4 (CH), 119.0 (C), 55.4 (CH_2_).; HREI-MS: *m*/*z* calcd for C_22_H_16_N_6_O_3_ [M]^+^ 400.1284; Found; 400.1135.

### 3.4. α-Glucosidase Inhibition Assay

To find out the α-glucosidase inhibition, kinetic studies were carried out using different concentrations of the inhibitors (0.0624, 0.3, 0.125, 0.4 mM) and different concentrations of substrate inhibitor (p-nitrophenol R-D maltoside NMP). Samples of 0.1, 0.2, 0.4, 0.8, and 1.0 mM (0.2 mg/mL) concentration were prepared in de-ionized water with the enzyme. PIPES buffer was used to adjust the pH of the solution. This solution was incubated for 30 min at 25 °C and the absorbance was recorded on an ELISA reader in 96-well plates. Kinetic parameters like Vmax, AICs, Km, and R2 were calculated using sigma plot enzyme kinetic software [24,25,26,27].

### 3.5. Molecular Docking Protocol

The molecular docking study was performed using discovery studio visualizer (DSV) and autodoc tools 1.5.7. The synthesized compounds were docked against α-glucosidase and their structures were obtained from the protein data bank (PDB) by searching for codes such as **3w37**. Initially, the protein was prepared using DSV to maintain the structure by removing water and the structure of the protein, as well as that of the selected analog, was saved in PDB format. The structure of α-glucosidase was made open in autodoc tools by adding polar hydrogen to the protein as well as Kollman and Gasteiger charges. Ligand preparation was done by using a torsion tree to detect the root and it was saved in PDBQT format. A configuration file was generated along with the X, Y, and Z axes, and the protein structures were saved in PDBQT. In order to generate different poses of molecules with varied energies, a command prompt was employed. The top-ranked molecules were docked with protein PDBQT format in DSV [28]. 

## 4. Conclusions

Benzotriazole-based bis-Schiff base scaffolds (**1**–**20**) were synthesized and then investigated in vitro for *α*-glucosidase inhibitory potentials. All the synthetic analogs based on benzotriazole-based bis-Schiff base scaffolds were found to display outstanding inhibition profiles on screening against the *α*-glucosidase enzyme. The synthesized scaffolds showed a range of inhibition profiles, having IC_50_ values ranging from 1.10 ± 0.05 µM to 28.30 ± 0.60 µM when compared to acarbose as a standard drug (IC_50_ = 10.30 ± 0.20 µM). Among the series, fifteen scaffolds 1–3, 5, 6, 9–16, 18–20 were identified to be more potent than standard acarbose, while the five remaining scaffolds 4, 7, 8, 16, and 17 also showed potency against the *α*-glucosidase enzyme but were found to be less potent than standard acarbose. The structures of all the newly synthesized scaffolds were confirmed using different spectroscopic techniques such as HREI-MS and ^1^H- and ^13^C- NMR spectroscopy. To find a structure-activity relationship, molecular docking studies were carried out to understand the binding mode of the active inhibitors with the active sites of the enzyme and the results supported the experimental data.

## Data Availability

Data is contained within the article.

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
