# Peer review of "Synthesis, In Vitro α-Glucosidase Inhibitory Activity and Molecular Docking Study of New Benzotriazole-Based Bis-Schiff Base Derivatives"

_pharmaceuticals, 2022, doi:10.3390/ph16010017_

Round 1

Reviewer 1 Report

 I. Khan and co-workers submitted a manuscript intitled: Synthesis, in vitro α-glucosidase inhibitory activity and molecular docking study of new benzotriazole-based bis-schiff base derivatives.

As well reported in the manuscript of the authors, benzotriazole molecules are key precursors in designing and development of biologically active drugs and are known to have broad range of biological activities such as antibacterial, antifungal, antihistaminic, anti-adrenergic and analgesic, anti-cancer and anti-convulsant activities. In this context, in recent years, different benzotriazole-based molecules are developed to generate a new biologically active heterocycles and, for these, reasons, the arguments is very actual and investigated.

In particular, in this manuscript, benzotriazole-based bis-schiff base scaffolds are synthesized and then illustrated in vitro for α-glucosidase inhibitory potentials. All the synthetics analogues based on benzotriazole-based bis-schiff base scaffolds display outstanding inhibition profile on screening against α-glucosidase enzyme. Synthetics scaffolds showed varied range of inhibition profile having IC50 values ranging from 1.10 ±0.05μM to 28.30 488 ±0.60μM when compared to acarbose as standard drug (IC50 = 10.30 ±0.20μM). Among the series, fifteen scaffolds were identified to be more potent than standard acarbose drug. The structure of all the newly synthetics scaffolds are confirmed by using HREI-MS, 1H- and 13C- NMR spectroscopy. To find  structure-activity relationship, molecular docking studies were carried out to understand the binding mode of active inhibitors with active site of enzymes and results supported the experimental data.

Consequently, the manuscript is well structured and well organized but major and minor revisions are required before being published in pharmaceutics.

-major revisions:

1) The whole experimental part must be correlated by the supporting information containing all the NMR spectra of the obtained products, all the HREI-MS spectra and all the data relating to the molecular docking.

2) In this project, no molecule with a substituent on the benzotriazole cycle has been synthesized (table 1, ring A).  Please, I suggest to synthesize at least one molecule putting a methoxy group or a nitro group.

3) In Table 1, different substituents are reported in ring B but, for completeness, the following should also be reported: 2-OMe, 4-OMe and 2-NO2.

4) In Table 1, different substituents are reported in ring C but, for completeness, the following should also be reported: 2-NO2, 2-OH and 3-OH.

For minor revision, I have attached a pdf file with different notes.

Best Regards

Author Response

Reviewer 1

1) The whole experimental part must be correlated by the supporting information containing all the NMR spectra of the obtained products, all the HREI-MS spectra and all the data relating to the molecular docking.

Reply: The spectral data related to synthesized analogs were provided in supplementary information and further data related to docking studies was also incorporated in manuscript (docking section) as per king suggestion.

2) In this project, no molecule with a substituent on the benzotriazole cycle has been synthesized (table 1, ring A).  Please, I suggest to synthesize at least one molecule putting a methoxy group or a nitro group.

Reply: I appreciate the suggestion of the worthy reviewer, but at this time we are not able to perform this experiment due to shortage of time, however in future manuscript this useful suggestion will be done practically

3) In Table 1, different substituents are reported in ring B but, for completeness, the following should also be reported: 2-OMe, 4-OMe and 2-NO2.

Reply: appreciate the suggestion of the worthy reviewer, but at this time we are not able to perform this experiment due to shortage of time, however in future manuscript this useful suggestion will be done practically

4) In Table 1, different substituents are reported in ring C but, for completeness, the following should also be reported: 2-NO2, 2-OH and 3-OH.

Reply: appreciate the suggestion of the worthy reviewer, but at this time we are not able to perform this experiment due to shortage of time, however in future manuscript this useful suggestion will be done practically

Reviewer 2 Report

This article by Rehman et. al. has described the design, synthesis, α-glucosidase inhibitory activity and molecular docking study of benzotriazole-based bis-schiff bases

The article is well written and could be beneficial to the relevant community after considering following points

1)      A rational of benzotriazole moiety for glucosidase inhibition through the literature should be elaborated, currently only two references 22 & 23 are not enough.

2)      Moreover, authors could shed light on the incorporation of bis-Schiff base groups, why did they choose these? Is there any prior literature about the enhancement of glucosidase inhibition by incorporating Schiff-base group?

3)      Scheme 1 should contain the details of R1 and R2 groups, although they are provided in the table 1, but better to describe underneath the chemical equation.

4)      Authors claimed that all the synthesized compounds are new, if so they must provide their original spectra, especially HRMS (mol. Wt.) and NMR (structure) as in supporting information.

Author Response

Reviewer = 02

This article by Rehman et. al. has described the design, synthesis, α-glucosidase inhibitory activity and molecular docking study of benzotriazole-based bis-schiff bases

The article is well written and could be beneficial to the relevant community after considering following points

  • A rational of benzotriazole moiety for glucosidase inhibition through the literature should be elaborated, currently only two references 22 & 23 are not enough.

Reply: Further references related to rationalization of benzotriazole were also incorporated as per kind suggestion

  • Moreover, authors could shed light on the incorporation of bis-Schiff base groups, why did they choose these? Is there any prior literature about the enhancement of glucosidase inhibition by incorporating Schiff-base group?

Reply: The literature about enhancement of glucosidase inhibition by incorporating Schiff-base group was incorporated as per kind suggestion

  • Scheme 1 should contain the details of R1 and R2 groups, although they are provided in the table 1, but better to describe underneath the chemical equation.

Reply: The detail about R1 & R2 was incorporated in scheme 1 underneath the chemical equation as per kind suggestion

  • Authors claimed that all the synthesized compounds are new, if so they must provide their original spectra, especially HRMS (mol. Wt.) and NMR (structure) as in supporting information.

Reply: The spectral data of representative analogs were provided in supplementary information

Reviewer 3 Report

In the present manuscript, Rehman and co-workers reported the synthesis of bis-schiff bases as an important building block bearing the benzotriazole ring. These functionalized aza-heterocyclic compounds were successfully synthesized, and their structures were characterized. In addition, the authors carried out biological studies of this interesting family of organic compounds. These compounds display an outstanding inhibition profile on screening against the α-glucosidase enzyme, showing a varied range of inhibition profiles having IC50 values ranging from 1.10 ±0.05µM to 28.30±0.60µM when compared to acarbose as a standard drug (IC50 = 10.30 ±0.20µM). To find a structure-activity relationship, molecular docking studies were carried out to understand the binding mode of active inhibitors with the active site of enzymes and the results supported the experimental data. In my opinion, the work (topical and results) is interesting for chemists and researchers in the synthetic organic and medicinal chemistry fields, and the synthetic method is very simple and practical. The manuscript is organized, has acceptable writing, the experiments were conducted very well, the results are very promising, and various products were well characterized. However, several things could be improved in typography, punctuation, syntax, word choice, analysis of results (especially of the synthetic aspect and the respective experimental section), etc.

Therefore, I believe this paper is suitable for publication in Pharmaceuticals after major revisions and answers to the following questions, suggestions, and corrections:

IMPORTANT:

 -Authors must include the Supplementary Information of the manuscript, which contains copies of the NMR and HREIMS spectra, HPLC of the evaluated compounds, etc.

-Introduction. Paragraph 1, this one is too long.

-Result and discussion. Several paragraphs are too long.

-The discussion of the synthetic results needs to be revised. Likewise, the respective conceptualization should be improved in the introduction.

-In the experimental section, the characterization data of the synthesized compounds must include the appearance of each one of the compounds, if they are oily or solid, the color they possess, and the melting point, if applicable. It should also be indicated if the described compound is new or has already been reported in the literature to compare the new results with those already described.

-A checklist must be attached.

-INTRODUCTION. The hypothesis or background of the experimental work developed could be better conceptualized.

- The authors had to evaluate the synthesis using other substituted benzotriazole derivatives to obtain a better molecular diversity that leads to a better structural/biological activity relationship analysis.

- Data from the HREIMS analysis should be reviewed in detail because several of them were quite wrong, and additionally. In addition, the respective spectra must be shown in the Supplementary Information.

- As there are products that have already been reported in the literature, the respective comparison of the melting point should be made, mention if the characterization data correspond to those of the literature, and cite the article. Indicate which compounds are new, if any, and which are already reported.

-Improve 13C NMR analysis by including the carbon type (i.e., C, CH, CH2, CH3) in each signal. In addition, the respective spectra must be shown in the Supplementary Information.

Several letters should be in italics (in nomenclature, the letter H and d in 1H-benzo[d], the J of coupling constants, etc.) and numbers in superscript (1H, 13C, etc.) or subscript (CDCl3, DMSO-d6, etc.).

OTHER CORRECTIONS:

Abstract. Change the following: ‘bis-schiff’ to ‘bis-Schiff’; ‘synthetics analogues’ to ‘synthetic analogs’; ‘display outstanding’ to ‘display an outstanding’; ‘against α-glucosidase enzyme’ to ‘against the α-glucosidase enzyme’; ‘showed varied range’ to ‘showed a varied range’; ‘profile having’ to ‘profiles having’; ‘as standard drug’ to ‘as a standard drug’; ‘as standard drug’ to ‘as a standard drug’; ‘16 and 17 also showed’ to ‘16, and 17, also showed’; ‘synthetics scaffolds were’ to ‘synthetic scaffolds was’; ‘find structure-activity’ to ‘find a structure-activity’; ‘active site of’ to ‘active sites of’; ‘enzymes and results’ to ‘enzymes, and the results’; ‘by using different type of spectroscopic’ to ‘using different spectroscopic’.

Introduction. Change the following: ‘high health care costs’ to ‘high healthcare costs’; ‘nephropathy and impaired’ to ‘nephropathy, and impaired’; ‘’ to ‘’; ‘in the postprandial blood’ to ‘in postprandial blood’; ‘delaying and prolonging’ to ‘delaying, and prolonging’; ‘and development of biologically’ to ‘and developing biologically’; ‘have broad range’ to ‘have a broad range’; ‘anti-adrenergic and analgesic’ to ‘anti-adrenergic, and analgesic’; ‘base analogues’ to ‘base analogs’; ‘we have plan to’ to ‘we plan to’; ‘benzotriazole-bearing’ to ‘benzotriazole bearing’; ‘in current study’ to ‘in the current study’; ‘as a potent alpha-glucosidase inhibitors’ to ‘as potent alpha-glucosidase inhibitors’; ‘benzotriazolebearing’ to ‘benzotriazole-bearing’; ‘base analogues’ to ‘base analogs’; ‘and easy to travel in cell’ to ‘and be easy to travel in cells’; ‘through cell wall’ to ‘through a cell wall’.

Result and discussion. Change the following: ‘After completion of reaction’ to ‘After completion of the reaction’; ‘of substrate (III)’ to ‘of the substrate (III)’; ‘acid and resulting mixture’ to ‘acid, and the resulting mixture’; ‘In order to treat’ to ‘To treat’; ‘inhibition of enzyme’ to ‘the inhibition of enzymes’; ‘to medicinal chemist’ to ‘from medicinal chemists’; ‘the synthetics analogues’ to ‘the synthetic analogues’; ‘outstanding inhibition’ to ‘an outstanding inhibition’; ‘varied range’ to ‘a varied range’; ‘α-glucosidase enzyme’ to ‘the α-glucosidase enzyme’; ‘inhibition profile having’ to ‘inhibition profiles having’; ‘as standard drug’ to ‘as a standard drug’; ‘16 and 17 also’ to ’16, and 17 also’; ‘potentials of α-glucosidase enzyme.’ to ‘potentials of the α-glucosidase enzyme.’; ‘(1, 9, 10 and 13) that bears’ to ‘(1, 9, 10, and 13) bearing’; ‘against α-glucosidase’ to ‘against the α-glucosidase’; ‘at 3-position’ to ‘at the 3-position’; ‘of α-glucosidase enzyme’ to ‘of the α-glucosidase enzyme’; ‘also bear substituent’ to ‘also bears substituent’; ‘same nature’ to ‘the same nature’; ‘but slight difference’ to ‘a slight difference’; ‘attributed to alteration in position of’ to ‘attribute to alteration in the position of’; ‘analogues (9)’ to ‘analogs (9)’; ‘the both aryl’ to ‘both aryl’; ‘with active residue’ to ‘with the active residue’; ‘with active site’ to ‘with the active site’; ‘The compound (1)’ to ‘Compound (1)’; ‘potency that compound (9)’ to ‘potency than compound (9)’; ‘in inhibition profile of analogue (1)’ to ‘in the inhibition profile of analog (1)’; ‘to replacement of –NO2’ to ‘to the replacement of –NO2’; ‘that nature of’ to ‘that the nature of’; ‘against α-glucosidase’ to ‘against the α-glucosidase’; ‘It was seemed from SAR’ to ‘It seemed from SAR’; ‘result to decreased’ to ‘resulted in decreased’; ‘α-glucosidase’ to ‘the α-glucosidase’; ‘hindrance as a result of which’ to ‘hindrance because of which’; ‘synthetic analogues with active’ to ‘synthetic analogs with the active’; ‘activity become lowered’ to ‘the activity becomes lowered’; ‘the analogue’ to ‘the analog’; ‘at 4-position’ to ‘at the 4-position’; ‘at 3-position’ to ‘at the 3-position’; ‘better competitor’ to ‘a better competitor’; ‘of same nature but have’ to ‘of the same nature but has’; ‘α-glucosidase’ to ‘the α-glucosidase’; ‘of compound’ to ‘of the compound’; ‘to lack of’ to ‘to the lack of’; ‘is capable of causing’ to ‘can cause’; ‘active residue’ to ‘the active residue’; ‘at 3-position’ to ‘at the 3-position’; ‘provide strong’ to ‘proving strong’; ‘at 4-position’ to ‘at the 4-position’; ‘at 3-position’ to ‘at the 3-position’; ‘provide strong’ to ‘providing strong’; ‘with active residue’ to ‘with the active residue’; ‘with active residue’ to ‘with the active residue’; ‘(2, 11, 12 and 14) that holds’ to ‘(2, 11, 12, and 14) that hold’; ‘be potent inhibitor’ to ‘be a potent inhibitor’; ‘alpha-glucosidase’ to ‘the alpha-glucosidase’; ‘, but have different position on both side of’ to ‘but have a different position on both sides of’; ‘potentials was affected’ to ‘potentials were affected’; ‘was dropped’ to ‘was drops’; ‘of substituent also’ to ‘of the substituent also’; ‘greatly affect the’ to ‘greatly affects the’; ‘hold’ to ‘holds’; ‘at 3-position’ to ‘at the 3-position’; ‘at 3-position’ to ‘at the 3-position’; ‘same’ to ‘the same’; ‘at 3-’ to ‘at the 3-’; ‘moiety result to’ to ‘moiety resulted in’; ‘in case of’ to ‘in the case of’; ‘of substituent of’ to ‘of the substituent of’; ‘greater number’ to ‘greater numbers’; ‘with active residue’ to ‘with the active residue’; ‘’ to ‘’; ‘at 4-position’ to ‘at the 4-position’; ‘at 4-position’ to ‘at the 4-position’; ‘was emerged as better competitor’ to ‘emerged as a better competitor’; ‘with active residue’ to ‘with the active residue’; ‘have strong tendency’ to ‘has a strong tendency’; ‘with inner cavity’ to ‘with the inner cavity’; ‘decrease in alpha-glucosidase’ to a decrease in the alpha-glucosidase’; ‘in inhibition profile’ to ‘in the inhibition profile’; ‘to fact that’ to ‘to the fact that’; ‘was due to the fact that di-Cl’ to ‘was because di-Cl’; ‘through conventional hydrogen bond with’ to ‘through a conventional hydrogen bonds’; ‘further drops down by attachment of substituent’ to ‘further dropped down by attachment of a substituent’; ‘active part of’ to ‘the active part of’;  ‘at the 4-position’ to ‘at the 4-position’; ‘at nature of’ to ‘at the nature of’; ‘to be low’ to ‘to be a low’; ‘alpha-glucosidase’ to ‘the alpha-glucosidase’; ‘to attached bulky –Br moiety at 4-position of’ to ‘to the attached bulky –Br moiety at the 4-position of’; ‘at other aryl part ‘at ‘another aryl part’ to ‘group cause steric’; ‘group cause steric’ to ‘group causes steric’; ‘inhibitory potentials’ to ‘the inhibitory potentials’; ‘holds same nature’ to ‘hold the same nature’; ‘on position of’ to ‘on the position of’; ‘at 4-position’ to ‘at the 4-position’; ‘’ to ‘’; ‘other aryl part ‘C’’ to ‘another aryl part ‘C’’; ‘at 4-position’ to ‘at the 4-position’; ‘have –Br moiety’ to ‘has –Br moiety’; ‘inhibition’ to ‘the inhibition’; ‘on position and’ to ‘on the position and’; ‘benzotriazole-based’ to ‘the benzotriazole-based’; ‘that alteration’ to ‘that the alteration’; ‘synthetics analogs’ to ‘synthetics analogues’; ‘α-glucosidase enzyme are listed in table-1.’ to ‘the α-glucosidase enzyme are listed in Table 1.’; ‘It was appeared from’ to ‘It appeared from’; ‘inhibition are elated strongly to position, nature and’ to ‘inhibition is related strongly to position, nature, and’; ‘Scheme-1;Table 1’ to ‘Scheme 1, Table 1’; ‘out in order to observe’ to ‘out to observe’; ‘nature and number’ to ‘nature, and the number’; ‘nature and number’ to ‘nature, and number’; ‘α-glucosidase enzyme’ to ‘at the α-glucosidase enzyme’; ‘from detail PLI’ to ‘from a detailed PLI’; ‘established number of’ to ‘established a number of’; ‘‘a number of key’ to ‘several key’; in improvement of inhibition’ to ‘in the improvement of the inhibition’; ‘against targeted’ to ‘against the targeted’; ‘of second most’ to ‘of the second most’; ‘with active sit’ to ‘with the active sit’; ‘The high potentials of’ to ‘The high potentials of’; ‘both side of aryl’ to ‘both sides of aryl’; ‘of electronic density’ to ‘of the electronic density’; ‘electron deficient’ to ‘electron-deficient’; ‘with active residue’ to ‘with the active residue’; ‘Moreover, attached’ to ‘Moreover, the attached’; ‘active residue’ to ‘the active residues’;

Conclusion. Change the following: ‘All the synthetics analogues’ to ‘All the synthetic analogs based’; ‘display outstanding’ to ‘display an outstanding’; ‘α-glucosidase’ to ‘the α-glucosidase’; ‘varied range’ to ‘a varied range’; ‘profile having’ to ‘profiles having’; ‘as standard’ to ‘as a standard’; ‘16 and 17 also’ to ’16, and 17 also’; ‘newly synthetics scaffolds were confirmed’ to ‘newly synthesized scaffolds was confirmed’; ‘find structure-activity’ to ‘find a structure-activity’; ‘with active site of enzymes and results’ to ‘with the active site of enzymes and the results’. Finally, the endpoint in the conclusions is missing.

Author Response

Reviewer = 03

In the present manuscript, Rehman and co-workers reported the synthesis of bis-schiff bases as an important building block bearing the benzotriazole ring. These functionalized aza-heterocyclic compounds were successfully synthesized, and their structures were characterized. In addition, the authors carried out biological studies of this interesting family of organic compounds. These compounds display an outstanding inhibition profile on screening against the α-glucosidase enzyme, showing a varied range of inhibition profiles having IC50 values ranging from 1.10 ±0.05µM to 28.30±0.60µM when compared to acarbose as a standard drug (IC50 = 10.30 ±0.20µM). To find a structure-activity relationship, molecular docking studies were carried out to understand the binding mode of active inhibitors with the active site of enzymes and the results supported the experimental data. In my opinion, the work (topical and results) is interesting for chemists and researchers in the synthetic organic and medicinal chemistry fields, and the synthetic method is very simple and practical. The manuscript is organized, has acceptable writing, the experiments were conducted very well, the results are very promising, and various products were well characterized. However, several things could be improved in typography, punctuation, syntax, word choice, analysis of results (especially of the synthetic aspect and the respective experimental section), etc.

Therefore, I believe this paper is suitable for publication in Pharmaceuticals after major revisions and answers to the following questions, suggestions, and corrections:

IMPORTANT:

 -Authors must include the Supplementary Information of the manuscript, which contains copies of the NMR and HREIMS spectra, HPLC of the evaluated compounds, etc.

Reply: The spectral data of the representative analogs were provided in the supplementary information as per kind suggestion

-Introduction. Paragraph 1, this one is too long.

Reply: The long sentences of paragraph 1 in introduction was shortened as per kind suggestion

-Result and discussion. Several paragraphs are too long.

Reply: The long sentences of result and discussion was shortened as per kind suggestion

-The discussion of the synthetic results needs to be revised. Likewise, the respective conceptualization should be improved in the introduction.

Reply: According to the kind suggestion of the reviewer, changes has been incorporated

-In the experimental section, the characterization data of the synthesized compounds must include the appearance of each one of the compounds, if they are oily or solid, the color they possess, and the melting point, if applicable. It should also be indicated if the described compound is new or has already been reported in the literature to compare the new results with those already described.

Reply: The synthesized compounds have not been reported yet, therefore physical state and color along with their respective melting point was incorporated as per kind suggestion

-A checklist must be attached.

Reply: The checklist was attached as per kind suggestion

-INTRODUCTION. The hypothesis or background of the experimental work developed could be better conceptualized.

Reply: The background analysis of developed work was further improved by incorporating more relevant references as per kind suggestion

- The authors had to evaluate the synthesis using other substituted benzotriazole derivatives to obtain a better molecular diversity that leads to a better structural/biological activity relationship analysis.

Reply: Dr Wajid Rehman sb

- Data from the HREIMS analysis should be reviewed in detail because several of them were quite wrong, and additionally. In addition, the respective spectra must be shown in the Supplementary Information.

Reply: Dr Wajid Rehman sb

- As there are products that have already been reported in the literature, the respective comparison of the melting point should be made, mention if the characterization data correspond to those of the literature, and cite the article. Indicate which compounds are new, if any, and which are already reported.

Reply: The synthesized compounds were not reported yet and further their respective melting point was incorporated as per kind suggestion 

-Improve 13C NMR analysis by including the carbon type (i.e., C, CH, CH2, CH3) in each signal. In addition, the respective spectra must be shown in the Supplementary Information.

Reply: The 13C NMR analysis was improved by including the carbon type (i.e., C, CH, CH2, CH3) in each signal and further spectral analysis of representative compounds were incorporated in supplementary information as per kind suggestion

Several letters should be in italics (in nomenclature, the letter H and d in 1H-benzo[d], the J of coupling constants, etc.) and numbers in superscript (1H, 13C, etc.) or subscript (CDCl3, DMSO-d6, etc.).

Reply: The above mentioned changes were incorporated as per kind suggestion

OTHER CORRECTIONS:

Abstract. Change the following: ‘bis-schiff’ to ‘bis-Schiff’;

Reply: The above mentioned change was incorporated as per kind suggestion

‘synthetics analogues’ to ‘synthetic analogs’;

Reply: The above mentioned change was incorporated as per kind suggestion

 ‘display outstanding’ to ‘display an outstanding’;

Reply: The above mentioned change was incorporated as per kind suggestion

 ‘against α-glucosidase enzyme’ to ‘against the α-glucosidase enzyme’;

Reply: The above mentioned change was incorporated as per kind suggestion

‘showed varied range’ to ‘showed a varied range’;

Reply: The above mentioned change was incorporated as per kind suggestion

‘profile having’ to ‘profiles having’;

Reply: The above mentioned change was incorporated as per kind suggestion

‘as standard drug’ to ‘as a standard drug’; ‘as standard drug’ to ‘as a standard drug’;

Reply: The above mentioned change was incorporated as per kind suggestion

‘16 and 17 also showed’ to ‘16, and 17, also showed’;

Reply: The above mentioned change was incorporated as per kind suggestion

‘synthetics scaffolds were’ to ‘synthetic scaffolds was’;

Reply: The above mentioned change was incorporated as per kind suggestion

‘find structure-activity’ to ‘find a structure-activity’;

Reply: The above mentioned change was incorporated as per kind suggestion

‘active site of’ to ‘active sites of’;

Reply: The above mentioned change was incorporated as per kind suggestion

‘enzymes and results’ to ‘enzymes, and the results’;

Reply: The above mentioned change was incorporated as per kind suggestion

‘by using different type of spectroscopic’ to ‘using different spectroscopic’.

Reply: The above mentioned change was incorporated as per kind suggestion

Introduction. Change the following: ‘high health care costs’ to ‘high healthcare costs’;

Reply: The above mentioned change was incorporated as per kind suggestion

 ‘nephropathy and impaired’ to ‘nephropathy, and impaired’;

Reply: The above mentioned change was incorporated as per kind suggestion

‘’ to ‘’; ‘in the postprandial blood’ to ‘in postprandial blood’;

Reply: The above mentioned change was incorporated as per kind suggestion

‘delaying and prolonging’ to ‘delaying, and prolonging’;

Reply: The above mentioned change was incorporated as per kind suggestion

‘and development of biologically’ to ‘and developing biologically’;

Reply: The above mentioned change was incorporated as per kind suggestion

‘have broad range’ to ‘have a broad range’;

Reply: The above mentioned change was incorporated as per kind suggestion

‘anti-adrenergic and analgesic’ to ‘anti-adrenergic, and analgesic’;

Reply: The above mentioned change was incorporated as per kind suggestion

‘base analogues’ to ‘base analogs’;

Reply: The above mentioned change was incorporated as per kind suggestion

‘we have plan to’ to ‘we plan to’; ‘benzotriazole-bearing’ to ‘benzotriazole bearing’;

Reply: The above mentioned change was incorporated as per kind suggestion

‘in current study’ to ‘in the current study’;

Reply: The above mentioned change was incorporated as per kind suggestion

‘as a potent alpha-glucosidase inhibitors’ to ‘as potent alpha-glucosidase inhibitors’;

Reply: The above mentioned change was incorporated as per kind suggestion

 ‘benzotriazolebearing’ to ‘benzotriazole-bearing’;

Reply: The above mentioned change was incorporated as per kind suggestion

‘base analogues’ to ‘base analogs’;

Reply: The above mentioned change was incorporated as per kind suggestion

‘and easy to travel in cell’ to ‘and be easy to travel in cells’;

Reply: The above mentioned change was incorporated as per kind suggestion

‘through cell wall’ to ‘through a cell wall’.

Reply: The above mentioned change was incorporated as per kind suggestion

Result and discussion. Change the following: ‘After completion of reaction’ to ‘After completion of the reaction’;

Reply: The above mentioned change was incorporated as per kind suggestion

 ‘of substrate (III)’ to ‘of the substrate (III)’;

Reply: The above mentioned change was incorporated as per kind suggestion

Reply: The above mentioned change was incorporated as per kind suggestion

 ‘acid and resulting mixture’ to ‘acid, and the resulting mixture’;

Reply: The above mentioned change was incorporated as per kind suggestion

 ‘In order to treat’ to ‘To treat’; ‘inhibition of enzyme’ to ‘the inhibition of enzymes’;

Reply: The above mentioned change was incorporated as per kind suggestion

 ‘to medicinal chemist’ to ‘from medicinal chemists’;

Reply: The above mentioned change was incorporated as per kind suggestion

 ‘the synthetics analogues’ to ‘the synthetic analogues’;

Reply: The above mentioned change was incorporated as per kind suggestion

 ‘outstanding inhibition’ to ‘an outstanding inhibition’;

Reply: The above mentioned change was incorporated as per kind suggestion

 ‘varied range’ to ‘a varied range’; ‘α-glucosidase enzyme’ to ‘the α-glucosidase enzyme’;

Reply: The above mentioned change was incorporated as per kind suggestion

 ‘inhibition profile having’ to ‘inhibition profiles having’;

Reply: The above mentioned change was incorporated as per kind suggestion

 ‘as standard drug’ to ‘as a standard drug’;

Reply: The above mentioned change was incorporated as per kind suggestion

 ‘16 and 17 also’ to ’16, and 17 also’;

‘potentials of α-glucosidase enzyme.’ to ‘potentials of the α-glucosidase enzyme.’;

Reply: The above mentioned change was incorporated as per kind suggestion

 ‘(1, 9, 10 and 13) that bears’ to ‘(1, 9, 10, and 13) bearing’;

Reply: The above mentioned change was incorporated as per kind suggestion

 ‘against α-glucosidase’ to ‘against the α-glucosidase’; ‘at 3-position’ to ‘at the 3-position’;

Reply: The above mentioned change was incorporated as per kind suggestion

 ‘of α-glucosidase enzyme’ to ‘of the α-glucosidase enzyme’;

Reply: The above mentioned change was incorporated as per kind suggestion

‘also bear substituent’ to ‘also bears substituent’;

Reply: The above mentioned change was incorporated as per kind suggestion

 ‘same nature’ to ‘the same nature’;

Reply: The above mentioned change was incorporated as per kind suggestion

 ‘but slight difference’ to ‘a slight difference’;

Reply: The above mentioned change was incorporated as per kind suggestion

 ‘attributed to alteration in position of’ to ‘attribute to alteration in the position of’;

Reply: The above mentioned change was incorporated as per kind suggestion

 ‘analogues (9)’ to ‘analogs (9)’;

Reply: The above mentioned change was incorporated as per kind suggestion

 ‘the both aryl’ to ‘both aryl’;

Reply: The above mentioned change was incorporated as per kind suggestion

 ‘with active residue’ to ‘with the active residue’;

Reply: The above mentioned change was incorporated as per kind suggestion

 ‘with active site’ to ‘with the active site’;

Reply: The above mentioned change was incorporated as per kind suggestion

 ‘The compound (1)’ to ‘Compound (1)’;

Reply: The above mentioned change was incorporated as per kind suggestion

 ‘potency that compound (9)’ to ‘potency than compound (9)’;

Reply: The above mentioned change was incorporated as per kind suggestion

 ‘in inhibition profile of analogue (1)’ to ‘in the inhibition profile of analog (1)’;

Reply: The above mentioned change was incorporated as per kind suggestion

 ‘to replacement of –NO2’ to ‘to the replacement of –NO2’;

Reply: The above mentioned change was incorporated as per kind suggestion

 ‘that nature of’ to ‘that the nature of’;

Reply: The above mentioned change was incorporated as per kind suggestion

 ‘against α-glucosidase’ to ‘against the α-glucosidase’;

Reply: The above mentioned change was incorporated as per kind suggestion

 ‘It was seemed from SAR’ to ‘It seemed from SAR’;

Reply: The above mentioned change was incorporated as per kind suggestion

 ‘result to decreased’ to ‘resulted in decreased’;

Reply: The above mentioned change was incorporated as per kind suggestion

 ‘α-glucosidase’ to ‘the α-glucosidase’;

Reply: The above mentioned change was incorporated as per kind suggestion

 ‘hindrance as a result of which’ to ‘hindrance because of which’;

Reply: The above mentioned change was incorporated as per kind suggestion

 ‘synthetic analogues with active’ to ‘synthetic analogs with the active’;

Reply: The above mentioned change was incorporated as per kind suggestion

 ‘activity become lowered’ to ‘the activity becomes lowered’;

Reply: The above mentioned change was incorporated as per kind suggestion

 ‘the analogue’ to ‘the analog’; ‘at 4-position’ to ‘at the 4-position’;

Reply: The above mentioned change was incorporated as per kind suggestion

 ‘at 3-position’ to ‘at the 3-position’;

Reply: The above mentioned change was incorporated as per kind suggestion

 ‘better competitor’ to ‘a better competitor’;

Reply: The above mentioned change was incorporated as per kind suggestion

 ‘of same nature but have’ to ‘of the same nature but has’;

Reply: The above mentioned change was incorporated as per kind suggestion

 ‘α-glucosidase’ to ‘the α-glucosidase’; ‘of compound’ to ‘of the compound’;

Reply: The above mentioned change was incorporated as per kind suggestion

 ‘to lack of’ to ‘to the lack of’;

Reply: The above mentioned change was incorporated as per kind suggestion

 ‘is capable of causing’ to ‘can cause’;

Reply: The above mentioned change was incorporated as per kind suggestion

 ‘active residue’ to ‘the active residue’;

Reply: The above mentioned change was incorporated as per kind suggestion

 ‘at 3-position’ to ‘at the 3-position’;

Reply: The above mentioned change was incorporated as per kind suggestion

 ‘provide strong’ to ‘proving strong’; ‘at 4-position’ to ‘at the 4-position’;

Reply: The above mentioned change was incorporated as per kind suggestion

 ‘at 3-position’ to ‘at the 3-position’;

Reply: The above mentioned change was incorporated as per kind suggestion

 ‘provide strong’ to ‘providing strong’;

Reply: The above mentioned change was incorporated as per kind suggestion

 ‘with active residue’ to ‘with the active residue’;

Reply: The above mentioned change was incorporated as per kind suggestion

 ‘with active residue’ to ‘with the active residue’;

Reply: The above mentioned change was incorporated as per kind suggestion

 ‘(2, 11, 12 and 14) that holds’ to ‘(2, 11, 12, and 14) that hold’;

Reply: The above mentioned change was incorporated as per kind suggestion

 ‘be potent inhibitor’ to ‘be a potent inhibitor’;

Reply: The above mentioned change was incorporated as per kind suggestion

 ‘alpha-glucosidase’ to ‘the alpha-glucosidase’;

Reply: The above mentioned change was incorporated as per kind suggestion

 ‘, but have different position on both side of’ to ‘but have a different position on both sides of’;

Reply: The above mentioned change was incorporated as per kind suggestion

 ‘potentials was affected’ to ‘potentials were affected’;

Reply: The above mentioned change was incorporated as per kind suggestion

 ‘was dropped’ to ‘was drops’;

Reply: The above mentioned change was incorporated as per kind suggestion

 ‘of substituent also’ to ‘of the substituent also’;

Reply: The above mentioned change was incorporated as per kind suggestion

 ‘greatly affect the’ to ‘greatly affects the’;

Reply: The above mentioned change was incorporated as per kind suggestion

 ‘hold’ to ‘holds’;

Reply: The above mentioned change was incorporated as per kind suggestion

 ‘at 3-position’ to ‘at the 3-position’;

Reply: The above mentioned change was incorporated as per kind suggestion

 ‘at 3-position’ to ‘at the 3-position’;

Reply: The above mentioned change was incorporated as per kind suggestion

 ‘same’ to ‘the same’;

Reply: The above mentioned change was incorporated as per kind suggestion

 ‘at 3-’ to ‘at the 3-’;

Reply: The above mentioned change was incorporated as per kind suggestion

 ‘moiety result to’ to ‘moiety resulted in’;

Reply: The above mentioned change was incorporated as per kind suggestion

 ‘in case of’ to ‘in the case of’;

Reply: The above mentioned change was incorporated as per kind suggestion

 ‘of substituent of’ to ‘of the substituent of’;

Reply: The above mentioned change was incorporated as per kind suggestion

 ‘greater number’ to ‘greater numbers’;

Reply: The above mentioned change was incorporated as per kind suggestion

 ‘with active residue’ to ‘with the active residue’;

 Reply: The above mentioned change was incorporated as per kind suggestion

 ‘’ to ‘’; ‘at 4-position’ to ‘at the 4-position’;

Reply: The above mentioned change was incorporated as per kind suggestion

 ‘at 4-position’ to ‘at the 4-position’;

Reply: The above mentioned change was incorporated as per kind suggestion

 ‘was emerged as better competitor’ to ‘emerged as a better competitor’;

Reply: The above mentioned change was incorporated as per kind suggestion

 ‘with active residue’ to ‘with the active residue’;

Reply: The above mentioned change was incorporated as per kind suggestion

 ‘have strong tendency’ to ‘has a strong tendency’;

Reply: The above mentioned change was incorporated as per kind suggestion

 ‘with inner cavity’ to ‘with the inner cavity’;

Reply: The above mentioned change was incorporated as per kind suggestion

 ‘decrease in alpha-glucosidase’ to a decrease in the alpha-glucosidase’;

Reply: The above mentioned change was incorporated as per kind suggestion

 ‘in inhibition profile’ to ‘in the inhibition profile’;

Reply: The above mentioned change was incorporated as per kind suggestion

 ‘to fact that’ to ‘to the fact that’;

Reply: The above mentioned change was incorporated as per kind suggestion

 ‘was due to the fact that di-Cl’ to ‘was because di-Cl’;

Reply: The above mentioned change was incorporated as per kind suggestion

 ‘through conventional hydrogen bond with’ to ‘through a conventional hydrogen bonds’;

Reply: The above mentioned change was incorporated as per kind suggestion

 ‘further drops down by attachment of substituent’ to ‘further dropped down by attachment of a substituent’;

Reply: The above mentioned change was incorporated as per kind suggestion

 ‘active part of’ to ‘the active part of’; 

Reply: The above mentioned change was incorporated as per kind suggestion

 ‘at the 4-position’ to ‘at the 4-position’; ‘at nature of’ to ‘at the nature of’;

Reply: The above mentioned change was incorporated as per kind suggestion

 ‘to be low’ to ‘to be a low’;

Reply: The above mentioned change was incorporated as per kind suggestion

 ‘alpha-glucosidase’ to ‘the alpha-glucosidase’;

Reply: The above mentioned change was incorporated as per kind suggestion

 ‘to attached bulky –Br moiety at 4-position of’ to ‘to the attached bulky –Br moiety at the 4-position of’;

Reply: The above mentioned change was incorporated as per kind suggestion

 ‘at other aryl part ‘at ‘another aryl part’ to ‘group cause steric’;

Reply: The above mentioned change was incorporated as per kind suggestion

 ‘group cause steric’ to ‘group causes steric’;

Reply: The above mentioned change was incorporated as per kind suggestion

 ‘inhibitory potentials’ to ‘the inhibitory potentials’;

Reply: The above mentioned change was incorporated as per kind suggestion

 ‘holds same nature’ to ‘hold the same nature’;

Reply: The above mentioned change was incorporated as per kind suggestion

 ‘on position of’ to ‘on the position of’;

Reply: The above mentioned change was incorporated as per kind suggestion

 ‘at 4-position’ to ‘at the 4-position’;

Reply: The above mentioned change was incorporated as per kind suggestion

 ‘’ to ‘’; ‘other aryl part ‘C’’ to ‘another aryl part ‘C’’;

Reply: The above mentioned change was incorporated as per kind suggestion

 ‘at 4-position’ to ‘at the 4-position’;

Reply: The above mentioned change was incorporated as per kind suggestion

 ‘have –Br moiety’ to ‘has –Br moiety’;

Reply: The above mentioned change was incorporated as per kind suggestion

 ‘inhibition’ to ‘the inhibition’;

Reply: The above mentioned change was incorporated as per kind suggestion

 ‘on position and’ to ‘on the position and’;

Reply: The above mentioned change was incorporated as per kind suggestion

 ‘benzotriazole-based’ to ‘the benzotriazole-based’;

Reply: The above mentioned change was incorporated as per kind suggestion

 ‘that alteration’ to ‘that the alteration’;

Reply: The above mentioned change was incorporated as per kind suggestion

 ‘synthetics analogs’ to ‘synthetics analogues’;

Reply: The above mentioned change was incorporated as per kind suggestion

 ‘α-glucosidase enzyme are listed in table-1.’ to ‘the α-glucosidase enzyme are listed in Table 1.’;

Reply: The above mentioned change was incorporated as per kind suggestion

 ‘It was appeared from’ to ‘It appeared from’;

Reply: The above mentioned change was incorporated as per kind suggestion

 ‘inhibition are elated strongly to position, nature and’ to ‘inhibition is related strongly to position, nature, and’;

Reply: The above mentioned change was incorporated as per kind suggestion

 ‘Scheme-1;Table 1’ to ‘Scheme 1, Table 1’;

Reply: The above mentioned change was incorporated as per kind suggestion

 ‘out in order to observe’ to ‘out to observe’;

Reply: The above mentioned change was incorporated as per kind suggestion

 ‘nature and number’ to ‘nature, and the number’;

Reply: The above mentioned change was incorporated as per kind suggestion

 ‘nature and number’ to ‘nature, and number’;

Reply: The above mentioned change was incorporated as per kind suggestion

 ‘α-glucosidase enzyme’ to ‘at the α-glucosidase enzyme’;

Reply: The above mentioned change was incorporated as per kind suggestion

 ‘from detail PLI’ to ‘from a detailed PLI’;

Reply: The above mentioned change was incorporated as per kind suggestion

 ‘established number of’ to ‘established a number of’;

Reply: The above mentioned change was incorporated as per kind suggestion

 ‘‘a number of key’ to ‘several key’; in improvement of inhibition’ to ‘in the improvement of the inhibition’;

Reply: The above mentioned change was incorporated as per kind suggestion

 ‘against targeted’ to ‘against the targeted’;

Reply: The above mentioned change was incorporated as per kind suggestion

 ‘of second most’ to ‘of the second most’;

Reply: The above mentioned change was incorporated as per kind suggestion

 ‘with active sit’ to ‘with the active sit’;

Reply: The above mentioned change was incorporated as per kind suggestion

 ‘The high potentials of’ to ‘The high potentials of’;

Reply: The above mentioned change was incorporated as per kind suggestion

 ‘both side of aryl’ to ‘both sides of aryl’;

Reply: The above mentioned change was incorporated as per kind suggestion

 ‘of electronic density’ to ‘of the electronic density’;

Reply: The above mentioned change was incorporated as per kind suggestion

 ‘electron deficient’ to ‘electron-deficient’;

Reply: The above mentioned change was incorporated as per kind suggestion

 ‘with active residue’ to ‘with the active residue’;

Reply: The above mentioned change was incorporated as per kind suggestion

 ‘Moreover, attached’ to ‘Moreover, the attached’; ‘active residue’ to ‘the active residues’;

Reply: The above mentioned change was incorporated as per kind suggestion

Conclusion. Change the following: ‘All the synthetics analogues’ to ‘All the synthetic analogs based’;

Reply: The above mentioned change was incorporated as per kind suggestion

‘display outstanding’ to ‘display an outstanding’;

Reply: The above mentioned change was incorporated as per kind suggestion

‘α-glucosidase’ to ‘the α-glucosidase’;

Reply: The above mentioned change was incorporated as per kind suggestion

‘varied range’ to ‘a varied range’;

Reply: The above mentioned change was incorporated as per kind suggestion

‘profile having’ to ‘profiles having’;

Reply: The above mentioned change was incorporated as per kind suggestion

‘as standard’ to ‘as a standard’;

Reply: The above mentioned change was incorporated as per kind suggestion

‘16 and 17 also’ to ’16, and 17 also’;

Reply: The above mentioned change was incorporated as per kind suggestion

‘newly synthetics scaffolds were confirmed’ to ‘newly synthesized scaffolds was confirmed’;

Reply: The above mentioned change was incorporated as per kind suggestion

‘find structure-activity’ to ‘find a structure-activity’;

Reply: The above mentioned change was incorporated as per kind suggestion

‘with active site of enzymes and results’ to ‘with the active site of enzymes and the results’.

Reply: The above mentioned change was incorporated as per kind suggestion

Finally, the endpoint in the conclusions is missing.

Reply: The above mentioned change was incorporated as per kind suggestion

Round 2

Reviewer 1 Report

I agree with the publication of the manuscript in this form.  if possible, I suggest to the authors to insert the references that I had indicated in the previous comments.

Best Regards

Author Response

Thanks for your valuable comments and acceptance of the changes we made and thanks for recommending our manuscript, the insertion of references at this stage may disturb the sequence and flow of manuscript however in future manuscript the same references may be included

Regards

Reviewer 2 Report

The authors have incorporated the suggested recommendations in the revised manuscript.

Author Response

Thanks for accepting the changes we made and thanks for recommending our article. We have double check the English grammatical and spell mistakes and make them correct

Regards

Reviewer 3 Report

The revised manuscript delivered almost all feedback in the right way. Now it's good enough to be accepted for publication in Pharmaceuticals after significant revisions in characterization data.

-Why didn't the authors attach supporting information?

-The HRMS analysis is still wrong for all compounds. In fact, in the new version, the error is even greater. For example, taking into account the data found for compounds 1, 2, and 3, the data should be reported as follows:

Compound 1: HREI-MS: m/z calcd for C22H17Cl2N5O2 [M] 453.0759 (or C22H18Cl2N5O2 [M + H]+ 454.0832); Found; 453.0630.

Compound 2: HREI-MS: m/z calcd for C22H18N6O3 [M] 414.1440 (or C22H19N6O3 [M + H]+ 415.1513); Found; 414.1070

Compound 3: HREI-MS: m/z calcd for C21H15BrN6O2 [M] 462.0440 (or C21H16BrN6O2 [M + H]+ 463.0518); Found; 462.0213.

...

A publishable HRMS analysis should have this math operation, calcd - found/calcd, with a value less than 5 ppm (or 5x10-6); the data found should vary approximately only in the last decimal place (the fourth, plus or minus 0.000x).

Finally, all the HRMS data is crucial because it is a fundamental part of the structural elucidation. Therefore, it would be convenient for them to consult an expert in the area to improve the analysis since the work should not be published with the condition of these results.  Should the respective experiments be repeated?

Author Response

The  mentioned changes have been incorporated as per kind suggestion in the manuscript.

Round 3

Reviewer 3 Report

The revised manuscript delivered almost all feedback. Now it's good enough to be accepted for publication in Pharmaceuticals.